# Stochastic Neural Ray Tracing for Radio Frequency Channel Modeling

Yinyan Bu [*1]   Jiajie Yu [*1]   Xingyu Chen [1]   Bo Wen [1]   Xinyu Zhang [1]   Piya Pal [1]

## Abstract

Wireless channel modeling is essential for the design, analysis, and optimization of modern wireless sensing and communication systems. However, accurately modeling wireless channel in electrically large and complex environments remains a long-standing challenge, owing to the intricate interactions between radio-frequency (RF) signals and surrounding objects (e.g., reflection, diffraction, and scattering). Unlike conventional ray-tracing pipelines that rely on hand-engineer interaction rules, or black-box neural surrogates that do not explicitly model physical structure, we introduce SNRFT, a novel framework that integrates neural representations with physics-based RF propagation modeling. Our key idea is to view RF transport as a stochastic propagation process, from which a material-dependent attenuation coefficient emerges naturally as the rate parameter governing transport dynamics. This formulation inherently satisfies key physical constraints such as reciprocity and reversibility. Building on this foundation, we employ implicit neural representations to capture complex RF-object interactions while preserving the composability of traditional ray tracing. Extensive evaluations on real-world wireless communication and sensing testbeds demonstrate that SNRFT consistently outperforms existing methods, while requiring significantly fewer training samples. Our code is available at: https://github.com/YinyanBu/SNRFT.

## 1. Introduction

Modern communication and sensing increasingly rely on wireless technologies that exploit electromagnetic (EM) waves for information exchange, fueling rapid advancements in mobile devices, automotive systems, and Internet-of-Things applications (De Alwis et al., 2021). Central to these developments is *wireless channel modeling*, whose accurate characterization is critical not only for optimizing the deployment of wireless network infrastructures (Wei et al., 2017) and designing spatial protection zones for spectrum sharing (Testolina et al., 2024), but also for enhancing the performance of wireless sensing applications, including object detection, localization, and imaging (Hu et al., 2023; Zhao et al., 2022; Vakalis et al., 2019; Bu et al., 2025). Although EM wave propagation is fundamentally governed by Maxwell's equations (Haus & Melcher, 1989), obtaining analytical or numerical solutions in realistic environments remains notoriously challenging due to complex geometries and nontrivial boundary conditions. These difficulties have motivated the development of a broad range of wireless channel modeling methodologies, which are commonly categorized into probabilistic models, deterministic models, and neural models.

Probabilistic models rely on statistical formulations to characterize wireless channels, typically estimating received signal strength as a function of the transmitter-receiver (TX-RX) separation. These models are grounded in empirical path-loss formulas whose parameters are calibrated through measurements collected in typical scenarios (Sarkar et al., 2003). However, they often lack accuracy and struggle to provide detailed channel characteristics. To address these limitations, deterministic models such as ray tracing algorithms have been extensively adopted in the wireless industry for environments with known 3D scene models (Remcom, 2024; Aoudia et al., 2025; Chen & Zhang, 2023; Chen et al., 2026). These methods represent EM waves as a dense collection of rays emitted from a TX, which interact with surrounding objects and are subsequently captured by RX. Despite their utility, conventional ray-tracing methods suffer from the inability to model *internal* or spatially varying structures. Furthermore, deterministic models are generally incompatible with probabilistic frameworks which have the advantage of better dealing with epistemic uncertainties.

Recent advances in neural rendering, including Neural Radiance Fields (Mildenhall et al., 2021) and 3D Gaussian Splatting (Kerbl et al., 2023), have demonstrated remarkable capabilities in capturing complex light transport for

---

[1]Department of Electrical and Computer Engineering, University of California San Diego (UCSD), La Jolla, United States. Correspondence to: Yinyan Bu <y1bu@ucsd.edu>, Jiajie Yu <jiy088@ucsd.edu>.

*Proceedings of the 43ʳᵈ International Conference on Machine Learning*, Seoul, South Korea. PMLR 306, 2026. Copyright 2026 by the author(s).

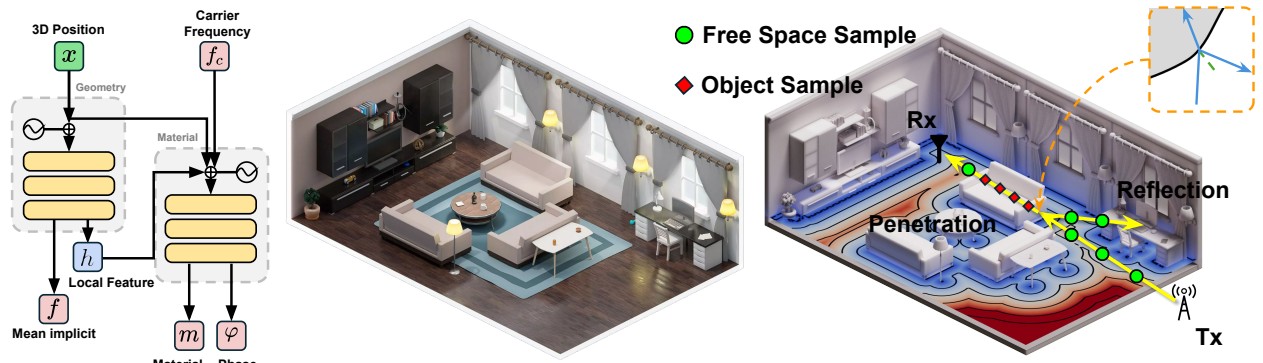

*Figure 1.* RF Ray Tracing with Neural Representation.

photorealistic scene reconstruction. These developments suggest promising avenues for RF propagation modeling, as both domains share underlying physical principles. Neural models, such as NeRF² (Zhao et al., 2023), NeWRF (Lu et al., 2024), RFGS (Yang et al., 2025) and WRF-GS (Wen et al., 2025), have shown potential to achieve high-fidelity in wireless data synthesis. Nevertheless, these methods often trade off the flexibility of conventional ray tracing and suffer from severe data inefficiency, requiring roughly 200 channel samples per square foot to achieve satisfactory performance (Chen et al., 2024). Moreover, their intermediate representations (i.e., attenuation) are purely black-box and lack physical interpretability.

In this paper, we propose **Stochastic Neural Radio Frequency Tracing** (SNRFT), a novel framework that synergistically combines the high fidelity of neural representations with the flexibility and interpretability of ray tracing as illustrated in Figure 1. Achieving this goal is challenging for several fundamental reasons. First, unlike vision systems, which use lenses to separate rays by direction, wireless systems receive superimposed RF signals at the antennas regardless of their propagation directions, making it difficult to associate measurements with specific propagation paths and complicating the integration of neural models with ray-tracing-based channel representations. Second, while visible light propagation is predominantly determined by scattering, RF reflections are dominated by specular reflections. Therefore, ray-tracing or ray-marching techniques that are designed for scattering-based vision sensors are unsuitable for RF wireless systems. Third, while modern cameras capture millions of spatial measurements per exposure, RF receivers typically contain only a single antenna or a small antenna array due to hardware constraints. The severe measurement sparsity requires physically informed inductive biases for tractable inference.

By addressing the above challenges, this paper makes the following key contributions:

- We propose SNRFT, a novel neural framework that bridges probabilistic modeling and physically grounded ray tracing, forming a unified and interpretable pipeline for wireless channel modeling.

- We model RF signal propagation between the TX and RX as a sequence of stochastic transport framework, and derive functional relationships linking attenuation, material properties, and stochastic object under physically grounded constraints.

- We evaluate SNRFT extensively across diverse real-world wireless environments. Our results demonstrate SNRFT's effectiveness and its potential to foster real-world wireless applications.

## 2. Background and Preliminaries

In this section, we provide background knowledge on RF ray tracing and wireless channel.

**RF Ray Tracing.** RF Ray tracing has emerged as a powerful technique for simulating radio wave propagation, driven by the rapid expansion of wireless communication and sensing applications. While full-wave EM solvers offer high-fidelity predictions, their prohibitive computational cost renders them impractical for large and electrically complex environments. In contrast, RF ray tracing leverages the principles of geometric optics to approximate EM wave behavior, providing a favorable trade-off between accuracy and computational efficiency (Ling et al., 1989; Chen et al., 2025; 2026). In this paradigm, RF waves are modeled as rays that interact with the environment through reflection, refraction, diffraction, and scattering.

The ray tracing process begins with environment modeling, where the physical scene is represented as a collection of object geometry, each annotated with electromagnetic material parameters such as relative permittivity and conductivity (Remcom, 2024). From each transmitter location, rays are launched in discrete angular directions with a prescribed

spatial resolution. Ray–surface intersections are then computed, and secondary rays are generated according to the number and types of interactions encountered. Depending on the interaction type (e.g., reflection, diffraction), ray trajectories are updated following geometric optics principles. At each interaction, the directional radiance (i.e., the power carried along a specific direction) is evaluated as a function of the material's electromagnetic properties. Ray propagation continues until a ray reaches the receiver or exceeds a predefined maximum number of interactions. Finally, all rays arriving at the RXs are coherently superimposed, with relative phases determined by their respective path lengths, to reconstruct the received signal.

**Wireless Channel.** A generic wireless communication system typically consists of TXs that generate and modulate information signals, which are then transmitted through the wireless channel to RXs. The transmitted signal can be represented as a complex-valued number $s_{\text{TX}} = Ae^{j\varphi}$, where $A$ denotes amplitude and $\varphi$ denotes phase respectively. During propagation, its amplitude is degraded by an attenuation factor $A_{\text{att}}$, and the phase is rotated by $\Delta\varphi$. In the case of *free-space* propagation (i.e., no obstacles obstruct the line-of-sight path), the amplitude attenuation is inversely proportional to the propagation distance, and the phase rotation is linearly inversely proportional to the distance (Molisch, 2012):

$$a(r) = \frac{c}{4\pi f_c r}, \quad \Delta\varphi(r) = -\frac{2\pi f_c r}{c}, \tag{1}$$

where $r$ is the propagation distance, $f_c$ is the carrier frequency, and $c$ is the speed of light. In realistic wireless environments, as the signal propagates through the scene, it encounters obstacles that cause reflections, diffraction, and scattering, resulting in multiple propagation paths. Consequently, the received signal can be expressed as (Tse & Viswanath, 2005)

$$y = Ae^{j\varphi} \sum_{l=1}^{L} T_l e^{j(2\pi f_c \tau_l + \Delta\varphi_l)}, \ \tau_l = \frac{d_l}{c}, \tag{2}$$

where $L$ is the total number of propagation paths, $T_l$ is the attenuated amplitude for path $l$, $\tau_l$ is the time delay of path length $d_l$, and $\Delta\varphi_l$ is the additional phase change from reflections or scatterings. The wireless channel $h$, is defined as the ratio of the received signal and transmitted signal:

$$h = \frac{y}{s} = \sum_{l=1}^{L} T_l e^{j(2\pi f_c \tau_l + \Delta\varphi_l)}. \tag{3}$$

## 3. Related Work

**RF and EM field Modeling.**

Conventional RF modeling methods include simulations (Remcom, 2024; Orekondy et al., 2023; MATLAB, 2025),

empirical models (Parsons, 2012; Hata, 2013), and physics-unaware deep-learning (DL) models (Liu et al., 2021; Malmirchegini & Mostofi, 2012), but all suffer from low modeling fidelity due to inherent limitations. Simulation-based methods rely on accurate Computer-Aided Design (CAD) models of the environment, which are often unavailable. Empirical models reduce complex propagation phenomena to a small set of parameters, resulting in oversimplified representations that can only predict coarse signal power. Physics-unaware DL models learn statistical mappings from data but fail to capture the underlying physical principles governing RF propagation.

Recent advances in scene representations have established a significant position in computer vision and computer graphics, particularly following the emergence of NeRF (Mildenhall et al., 2021) and 3D Gaussian Splatting (Kerbl et al., 2023). Several studies have adapted neural representations and 3D Gaussian models in computer vision for RF applications, with NeRF² (Zhao et al., 2023), NeWRF (Lu et al., 2024) and WRF-GS (Wen et al., 2025) as being notable examples. These methods represent the entire scene as either an implicit neural function or a collection of analytical 3D Gaussian kernels, which hinders explicit boundary delineation and the isolation of object-specific RF interaction properties. This limitation reduces adaptability to diverse RF environments and typically requires a larger volume of training data. More recently, RFscape (Chen et al., 2025) leverages a learnable signed distance function (SDF) alongside a neural material network to jointly model complex geometries and RF material properties within a ray tracing pipeline. However, it does not explicitly incorporate propagation-level physical constraints, such as reciprocity, which is a relevant structural property in certain wireless systems (e.g., time-division duplex (TDD) systems (Marzetta, 2010)). Explicitly accounting for such structure provides strong inductive biases and enables more sample-efficient RF channel modeling from sparse real-world measurements.

**Exponential Transport for Stochastic Object.** Previous work has explored exponential transport (see 4.2) models in the context of stochastic object for rendering colors. For example, Mishchenko et al. (2006) studied exponential transport for stochastic microparticle distributions, which can be formalized as a Poisson Boolean model of stochastic object (d'Eon, 2018; Jarabo et al., 2018), where particle locations are independent and modeled as a spatial Poisson process (Chiu et al., 2013). Within this framework, the attenuation coefficient can be derived analytically from the probability distributions governing particle location, size, material, shape, and orientation (Heitz et al., 2015; Jakob et al., 2010). More recently, Miller et al. (2024) derived volumetric representations of opaque solids directly from the axioms of exponential transport using stochastic theory, providing a principled explanation for why volumetric neural rendering

methods are able to recover solid object.

Building upon these insights, we advance the theory of stochastic object from optics into the RF domain, where several fundamental differences make the problem significantly more challenging. Unlike visible light, RF signals possess much longer wavelengths, which enable them to penetrate common materials. This distinction fundamentally invalidates the assumption in Miller et al. (2024) that rays are terminated at opaque solid boundaries, since in RF propagation, the dominant effect is cumulative attenuation within the objects. Moreover, attenuation in RF propagation is inherently material-dependent: different media (e.g., concrete, wood, metal) induce distinct propagation behaviors governed by material property such as permittivity, conductivity. In contrast, the formulation of Miller et al. (2024) is agnostic to material properties and is developed in the context of optical transport in free space, and therefore does not capture the physics of RF propagation (within objects). By explicitly incorporating object penetration and material-dependent attenuation representations, our formulation provides the first stochastic object framework that rigorously explains how RF signals propagate and interact with complex environments.

## 4. Methodology

### 4.1. Problem Formulation

Formally, given a TX located at $\boldsymbol{p}_{\text{TX}} = (x_{\text{TX}}, y_{\text{TX}}, z_{\text{TX}})$, the transmitted signal $s_{\text{TX}}$ and a set of RX positions $\{\boldsymbol{p}_{\text{RX}}^{(m)} = (x_{\text{RX}}^{(m)}, y_{\text{RX}}^{(m)}, z_{\text{RX}}^{(m)})\}_{m=1}^{M}$, the goal is to estimate a model with parameter $\boldsymbol{\Theta}$ that synthesizes the received complex-valued RF signal $s_{\text{RX}}^{(m)}$ at each $\boldsymbol{p}_{\text{RX}}^{(m)}$.

### 4.2. Definitions, Notations and Assumptions

While NeRF-based methods (Zhao et al., 2023; Lu et al., 2024; Chen et al., 2024) leverage voxel-based representations to capture scene impact on RF signal propagation and employ ray-marching-based rendering to achieve state-of-the-art fidelity in RF data synthesis, they all rely on a key underlying assumption:

**Assumption 4.1.** Ray propagation induces exponential attenuation of amplitude.

This behavior aligns with the physical model described in the ITU-R P.2040-1 recommendation, which states the *exponential* decrease of the electric field with distance. We refer to this assumption as *exponential transport*. However, these approaches overlook fundamental physical constraints of wireless signal propagation, such as channel reciprocity, which plays a central role in time-division duplexing (TDD) systems by enabling downlink channel inference from uplink measurements in practical 5G networks

(Marzetta, 2010). Motivated by this observation, we seek to incorporate physical constraints (e.g., reciprocity-aware) into RF signal propagation models.

To set the stage for rigorous analysis of exponential transport in RF ray tracing, we first introduce some notations and definitions for stochastic object (similar to those used in (Miller et al., 2024)) and object-inside propagation.

**Notations.** We use $\boldsymbol{r}_{\boldsymbol{x},\boldsymbol{\omega}}(t) \triangleq \boldsymbol{x} + t \cdot \boldsymbol{\omega}$ to denote the point on a ray with origin $\boldsymbol{x} \in \mathbb{R}^3$ and direction $\boldsymbol{\omega} \in \mathcal{S}^2$ after propagating distance $t \in [0, \infty)$. We denote $d_{\boldsymbol{x},\boldsymbol{\omega}}^*$ as the object-inside propagation distance and $\boldsymbol{r}_{\boldsymbol{x},\boldsymbol{\omega}}(d_{\boldsymbol{x},\boldsymbol{\omega}}^*)$ as the intersection point between object and free-space.

**Definition 4.2.** Define a indicator function $I : \mathbb{R}^3 \to \{0, 1\}$ as a binary scalar field, and associate it with a object $\mathcal{O} \triangleq \{\boldsymbol{x} \in \mathbb{R}^3 : I(\boldsymbol{x}) = 1\}$. When the indicator function $I(\boldsymbol{x})$ is a random scalar field, the associated $\mathcal{O}$ is called a stochastic object, for which we define the occupancy $o : \mathbb{R}^3 \to [0, 1]$ and vacancy $v : \mathbb{R}^3 \to [0, 1]$:

$$
\begin{aligned}
o(\boldsymbol{x}) &= \Pr(I(\boldsymbol{x}) = 1), \\
v(\boldsymbol{x}) &= \Pr(I(\boldsymbol{x}) = 0) = 1 - o(\boldsymbol{x}).
\end{aligned}
\tag{4}
$$

**Definition 4.3.** In a scene with stochastic object $\mathcal{O}$, define the tail distribution of the object-inside propagation distance $d_{\boldsymbol{x},\boldsymbol{\omega}}^*$:

$$
T_{\boldsymbol{x},\boldsymbol{\omega}}(t) \triangleq \Pr\{d_{\boldsymbol{x},\boldsymbol{\omega}}^* \geq t\}.
\tag{5}
$$

The object-inside propagation distribution is the probability density function (pdf) of object-inside propagation distance $d_{\boldsymbol{x},\boldsymbol{\omega}}^*$:

$$
p_{\boldsymbol{x},\boldsymbol{\omega}}(t) = \frac{\mathrm{d}(1 - T_{\boldsymbol{x},\boldsymbol{\omega}}(t))}{\mathrm{d}t} = -\frac{\mathrm{d}T_{\boldsymbol{x},\boldsymbol{\omega}}(t)}{\mathrm{d}t}.
\tag{6}
$$

The *absolute value/magnitude* of attenuation coefficient $a(\boldsymbol{x}, \boldsymbol{\omega})$ at point $\boldsymbol{x}$ and direction $\boldsymbol{\omega}$ within object is the pdf value of zero object-inside propagation distance:

$$
|a(\boldsymbol{x}, \boldsymbol{\omega})| = \sigma(\boldsymbol{x}, \boldsymbol{\omega}) = p_{\boldsymbol{x},\boldsymbol{\omega}}(0).
\tag{7}
$$

$T_{\boldsymbol{x},\boldsymbol{\omega}}$ inherits the following properties from ray propagation: (*i*) *monotonically non-increasing* since $T_{\boldsymbol{x},\boldsymbol{\omega}}(t) \leq T_{\boldsymbol{x},\boldsymbol{\omega}}(s)$ if $t > s$; (*ii*) $T_{\boldsymbol{x},\boldsymbol{\omega}}(0) = 1$.

### 4.3. Exponential Transport for RF Ray Propagation

Under Assumption 4.1, Equations (5) to (7) imply:

$$
T_{\boldsymbol{x},\boldsymbol{\omega}}(t) = \exp(-\int_0^t \sigma(\boldsymbol{r}_{\boldsymbol{x},\boldsymbol{\omega}}(s), \boldsymbol{\omega})\mathrm{d}s),
\tag{8}
$$

$$
p_{\boldsymbol{x},\boldsymbol{\omega}}(t) = \sigma(\boldsymbol{r}_{\boldsymbol{x},\boldsymbol{\omega}}(t), \boldsymbol{\omega})T_{\boldsymbol{x},\boldsymbol{\omega}}(t).
\tag{9}
$$

Therefore, the magnitude of attenuation coefficient stands for the exponential rate parameter of the propagation distance.

Since the wavelength of light ($\sim 10^{-7}$ m) is far smaller than typical surface irregularities, diffuse reflection is the primary outcome. However, RF signals, with wavelengths of order $10^{-3}$ m to $10^{-1}$ m, perceive most surfaces as smooth, and thus exhibit predominantly specular rather than diffuse reflections (Lu et al.). Therefore, we primarily focus on specular reflection and penetration in our theoretical framework. Each propagation path of the RF signals between the TX and the RX is modeled as a sequence of independent propagation segments, where each segment corresponds either to free-space propagation or object-inside propagation until reaching a boundary. Right after the intersection, we consider reflective and penetrative rays for the next ray propagation, which also implies for later intersection points. Before stating our main theoretical results, we emphasis the physical plausibility for ray tracing of RF signals:

($i$) **Basic reciprocity.** Consider two points $\boldsymbol{x}$, $\boldsymbol{y}$ both in free-space or inside-object, $T_{\boldsymbol{x},\boldsymbol{\omega}}(t) = T_{\boldsymbol{y},-\boldsymbol{\omega}}(t)$ if $\boldsymbol{y} = \boldsymbol{r}_{\boldsymbol{x},\boldsymbol{\omega}}(t)$.

($ii$) **Reversibility.** Object inferred from $I$ is consistent when evaluated along any ray, regardless of whether the ray direction $\boldsymbol{\omega}$ is forward or reversed.

($iii$) **Material-dependent attenuation.** RF signals attenuate at different rates in different media/material. For example, for a segment of length $t$ inside a *homogeneous* material $m$, we have
$$T_{\boldsymbol{x},\boldsymbol{\omega}}(t) = e^{-\sigma_m t},$$
where $\sigma_m > 0$ denotes the attenuation coefficient of material $m$, reflecting the physically observed fact that different media absorb RF energy at different rates.

It can be readily verified that, once the basic reciprocity condition holds, the reciprocity of an entire propagation path (e.g., multiple scattering events and successive penetrations through objects) is automatically ensured. We are now ready to state our main technical result.

**Theorem 4.4.** *Consider a random indicator function $I$ and associated stochastic object geometry $\mathcal{O}$. We assume that $\sigma(\boldsymbol{x},\boldsymbol{\omega})$ is differentiable with respect to point $\boldsymbol{x}$, with its gradient bounded for any $\boldsymbol{\omega} \in \mathcal{S}^2$. Then, for any ray with origin $\boldsymbol{x}$ in a certain object, direction $\boldsymbol{\omega}$, the distribution of the object-inside propagation distance $d^*_{\boldsymbol{x},\boldsymbol{\omega}}$ is exponential if and only if $I_{\boldsymbol{x},\boldsymbol{\omega}}$, constitutes a continuous-time discrete-state Markov process. Formally, this condition requires that for any $t$:*
$$\begin{aligned} &\Pr\{I_{\boldsymbol{x},\boldsymbol{\omega}}(t) \mid I_{\boldsymbol{x},\boldsymbol{\omega}}(t_n), t_n < t, n = 1, \ldots, N\} \\ &= \Pr\{I_{\boldsymbol{x},\boldsymbol{\omega}}(t) \mid I_{\boldsymbol{x},\boldsymbol{\omega}}(\max_n t_n)\}. \end{aligned} \tag{10}$$

*Furthermore, the process is physically plausible (i.e. satisfying reciprocity and reversibility) if the attenuation magnitude $\sigma(\boldsymbol{x},\boldsymbol{\omega})$ satisfies:*
$$\sigma(\boldsymbol{x},\boldsymbol{\omega}) = \frac{(m(\boldsymbol{x})(1 - v(\boldsymbol{x})) + 1)|\boldsymbol{\omega} \cdot \nabla v(\boldsymbol{x})|}{1 - v(\boldsymbol{x})}. \tag{11}$$

We provide the detailed discussion in Appendix D.

**Beyond binary object representation.** Definition 4.2 define object through the binary indicator function. In practice, it is often preferable to represent geometry via a non-binary scalar field, which can encode richer information (Osher et al., 2004). We consider a generalized definition of non-binary random field: an implicit function $G : \mathbb{R}^3 \to \mathbb{R}$ is defined as a random real scalar field with associated indicator $I(\boldsymbol{x}) = \mathbf{1}_{G(\boldsymbol{x})\leq 0}$, we define CDF, PDF and mean of random field $G$:
$$\begin{aligned} \mathrm{CDF}_{\boldsymbol{x}}(q) &\triangleq \Pr\{G(\boldsymbol{x}) \leq q\}, q \in \mathbb{R} \\ \mathrm{PDF}_{\boldsymbol{x}}(q) &\triangleq \frac{\mathrm{d}\,\mathrm{cdf}(q)}{\mathrm{d}q}, q \in \mathbb{R} \\ f(\boldsymbol{x}) &\triangleq \mathbb{E}[G(\boldsymbol{x})]. \end{aligned} \tag{12}$$

Following (Miller et al., 2024), we assume that $G(\boldsymbol{x}) = f(\boldsymbol{x}) + \frac{1}{s}Z(\boldsymbol{x})$, where $Z(\boldsymbol{x})$ is a zero-mean symmetric random field with CDF $\Psi(\cdot)$ and PDF $\psi(\cdot)$, then
$$\begin{aligned} v(\boldsymbol{x}) &= \Pr(G(\boldsymbol{x}) > 0) = \Psi(sf(\boldsymbol{x})), \\ o(\boldsymbol{x}) &= \Pr(G(\boldsymbol{x}) \leq 0) = \Psi(-sf(\boldsymbol{x})), \end{aligned} \tag{13}$$

we can further generalize the representation in Theorem 4.4:
$$\sigma(\boldsymbol{x},\boldsymbol{\omega}) = \frac{(m(\boldsymbol{x})\Psi(-sf(\boldsymbol{x})) + 1)s\psi(sf(\boldsymbol{x}))|\omega \cdot \nabla f(\boldsymbol{x})|}{\Psi(-sf(\boldsymbol{x}))}. \tag{14}$$
$Z(\boldsymbol{x})$ may follow any zero-mean, unit-variance symmetric distribution (e.g., Laplace, Gaussian, Logistic), $s > 0$ represents a pre-defined scaling coefficient.

### 4.4. RF Ray Tracing with SNRFT

We model RF signal propagation using a neural implicit representation embedded within a differentiable ray-tracing framework. To characterize stochastic scene geometry, we employ the implicit mean function $f(\boldsymbol{x})$ introduced in Section 4.3. However, geometry alone is insufficient for accurately modeling RF propagation, since the complex-valued attenuation coefficient also depends on local material properties and phase responses of the medium. Specifically, the attenuation coefficient is jointly determined by the spatially varying material parameters $m(\boldsymbol{x})$ and phase shift $\varphi(\boldsymbol{x})$. Conventional RF ray tracing methods (Ling et al., 1989) typically compute Fresnel coefficients under the assumption of homogeneous and fully known materials. In practice, however, real-world environments exhibit strong spatial heterogeneity across both object surfaces and interiors, while material properties are often inaccessible or difficult to measure directly. To address this limitation, we parameterize these unknown propagation characteristics using a trainable neural field. Inspired by NeRF (Mildenhall et al., 2021), we adopt multi-layer perceptrons (MLPs) with the following

mapping:

$$F_{\Theta} : (\boldsymbol{x}, f_c) \mapsto \big(f(\boldsymbol{x}), m(\boldsymbol{x}), \Delta\varphi(\boldsymbol{x})\big), \qquad (15)$$

where $\boldsymbol{x} \in \mathbb{R}^3$ denotes the spatial coordinate and $f_c$ is the RF carrier frequency. The resulting complex-valued attenuation coefficient is expressed as

$$a(\boldsymbol{x}, \boldsymbol{\omega}) = \sigma(\boldsymbol{x}, \boldsymbol{\omega}) \, e^{j\varphi(\boldsymbol{x})}, \qquad (16)$$

where $\sigma(\boldsymbol{x}, \boldsymbol{\omega})$ follows (14) when $\boldsymbol{x}$ lies inside an object, and reduces to the free-space propagation model in (1) otherwise.

Following standard RF ray-tracing procedures, rays are emitted from the transmitter (Tx) and recursively propagated through the scene. A key operation during propagation is identifying ray-object intersections. In our framework, intersections are determined by searching for zero-crossings of the implicit function $f(\boldsymbol{x})$ along the ray direction. The iterative search terminates once $|f(\boldsymbol{x})| < \epsilon$ or a predefined maximum depth is reached. At each intersection point, the local surface normal is estimated using finite differences:

$$[\boldsymbol{n}(\boldsymbol{x})]_j = \frac{f(\boldsymbol{x} + \epsilon_j) - f(\boldsymbol{x} - \epsilon_j)}{\|f(\boldsymbol{x} + \epsilon_j) - f(\boldsymbol{x} - \epsilon_j)\|}, \text{ for } j = 1, 2, 3. \qquad (17)$$

where $\boldsymbol{e}_j$ denotes the unit basis vector along the $j$-th coordinate axis. The direction of the specular reflected ray is then computed using specular reflection determined by Snell's law (Balanis, 2012) (i.e., $\boldsymbol{\omega}' = \boldsymbol{\omega} - 2\langle \boldsymbol{\omega}, \boldsymbol{n}\rangle \boldsymbol{n}$), while penetrative rays preserve their incident direction. To control the growth of ray branches, we adopt the Monte Carlo path selection strategy as in RFScape (Chen et al., 2025), where reflective and penetrative rays are probabilistically sampled according to their directional attenuation coefficients.

Ray interactions are recursively evaluated until valid propagation paths connecting the Tx and receiver (Rx) are identified. Each valid path $l$ consists of multiple propagation segments, where each segment lies either in free space or inside an object. We characterize each valid path $l$ using three quantities: the estimated accumulated transmittance $\hat{T}_l$, the estimated accumulated phase shift $\Delta\hat{\varphi}_l$, and the estimated total traversal distance $\hat{d}_l$. Specifically,

$$\hat{T}_l = \prod_i \exp\big(-\hat{a}_l^{(i)}\big) = \exp\big(-\sum_i \hat{a}_l^{(i)}\big), \qquad (18)$$

where $\hat{a}_l^{(i)}$ denotes the estimated accumulated attenuation along segment $i$. For segments inside an object, $\hat{a}_l^{(i)}$ is approximated by sampling along the ray with a small step size $\Delta t$:

$$\hat{a}_l^{(i)} = \int_0^{d_l^{(i)}} \hat{a}(\boldsymbol{p}_0 + t \cdot \boldsymbol{\omega}) \mathrm{d}t \approx \sum_{n=0}^{N_l^{(i)}-1} \hat{a}(\boldsymbol{p}_0 + n\Delta t \cdot \boldsymbol{\omega})\Delta t, \qquad (19)$$

where $\boldsymbol{p}_0$ denotes segment origin and $N_l^{(i)} = \frac{\hat{d}_l^{(i)}}{\Delta t}$ is the total number of samples along the ray segment. For free-space segments, $\hat{a}_l^{(i)}$ is computed analytically using (1): $\hat{a}_l^{(i)} = \frac{c}{4\pi f_c \hat{d}_l^{(i)}}$. The estimated accumulated phase shift is modeled as

$$\Delta\hat{\varphi}_l = \sum_i \Delta\hat{\varphi}_l^{(i)}, \qquad (20)$$

where $\Delta\hat{\varphi}l^{(i)}$ is predicted by evaluating the neural representation $F_{\Theta}$ at the corresponding segment endpoint. The estimated time fly for valid path $l$ is computed as

$$\hat{\tau}_l = \frac{\hat{d}_l}{c}. \qquad (21)$$

Finally, the estimated received RF signal is obtained by coherently aggregating contributions from all valid propagation paths:

$$\hat{s}_{\text{RX}} = s_{\text{TX}} \cdot \sum_l \hat{T}_l \cdot e^{j(2\pi f \hat{\tau}_l + \Delta\varphi_l)}. \qquad (22)$$

The entire propagation process is differentiable with respect to the neural field parameters $\Theta$, enabling end-to-end optimization directly from RF observations.

### 4.5. Optimization

To accurately model RF signal propagation, we optimize the parameters $\Theta$ by minimizing the sim-to-real discrepancy, i.e., $\sum_{m=1}^{M} \mathcal{L}(s_{\text{RX}}^{(m)}, \hat{s}_{\text{RX}}^{(m)})$, where $s_{\text{RX}}^{(m)}$ is $m$-th RF measurement in the scene, $M$ is the number of measurements, $\mathcal{L}$ is the loss that quantifies the discrepancy between real measurements and model prediction. The overall optimization objective becomes

$$\Theta^o = \arg\min_{\Theta} \sum_{m=1}^{M} \mathcal{L}(s_{\text{RX}}^{(m)}, \hat{s}_{\text{RX}}^{(m)}). \qquad (23)$$

To encourage smooth and physically consistent reconstructions, we additionally incorporate a discrete Laplacian regularization. We expect that only a small number of RF measurements (i.e., small $M$) is sufficient, since the governing physical laws of ray propagation provide strong structural constraints.

*Remark* 4.5. The signed distance function (SDF) can be regarded as a special case of $f(\boldsymbol{x})$. In this case, an additional eikonal regularization (Sethian, 1996) is required to enforce the unit-norm constraint on the SDF gradient.

## 5. Experiments

### 5.1. Model Implementation and Training

- Structure Model: we employ an 8-layer Multilayer Perceptron (MLP) with a hidden dimension of 64 and integration

of skip connections. The model takes 3D coordinates as input, i.e. $G_\Omega : \boldsymbol{x} \to (f, \boldsymbol{h})$, where an additional 128-dimensional local features $\boldsymbol{h}$ is the output of the geometry model. To cope with the spectral bias inherent in coordinate-based MLP, the 3D coordinate input $\boldsymbol{x}$ undergoes sinusoid positional embedding with 6 frequencies.

- Material Model: the material model consists of an 8-layer MLP with a width of 128 and incorporates skip connections, where the inputs are spatial coordinate $\boldsymbol{x}$, carrier frequency $f_c$ and feature $\boldsymbol{h}$ from structure model. This architecture is designed to capture electromagnetic material properties.

- The training configuration includes a batch size of 4 and utilizes the Adam optimizer (Kingma & Ba, 2014) with default momentum parameters $\beta_1 = 0.9$ and $\beta_2 = 0.999$. The learning rate is set at $1 \times 10^{-4}$ and is annealed to $1 \times 10^{-6}$ using a cosine schedule. On a NVIDIA RTX 5090 GPU, the network typically converges after approximately $20k$ iterations, which takes about 7 hours.

## 5.2. Room-Level BLE RSSI prediction

### 5.2.1. EXPERIMENTAL SETTINGS

**Task.** This task verifies that SNRFT supports single-antenna setups for capturing a single real-valued RSSI. Given a TX (BLE node) sending signals from location $(x_{\text{TX}}, y_{\text{TX}}, z_{\text{TX}})$, the goal is to synthesize the RSSI (in dB) received by a RX (BLE gateway with a single antenna). The measured RSSI represents the power of aggregate signal from all directions.

**Dataset.** We use an real-world open-source BLE dataset provided in NeRF[2] (Zhao et al., 2023), where the facility occupies $15000 \, \text{ft}^2$. It contains 21 BLE gateways, which operate at 2.4 GHz to collect the ID and RSSI of BLE beacons. Each dataset item is a 21-dimensional tuple, including the RSSI values detected by 21 gateways, plus the position of the BLE node. The RSSI value is set to -100 dB by default if the gateway does not detect any signal.

**Baselines.** We employ **NeRF**[2] (Zhao et al., 2023) as a baseline, as it represents the state-of-the-art in RSSI estimation, already showing superior performance to other deep learning based approaches such as CGAN (Parralejo et al., 2021). Moreover, we introduce two more baselines, **k-NN**: Predicting the channel given the closest match to the input spatial coordinates in terms of Euclidean distance; **MRI** (Shin et al., 2014): Interpolating the RSSI values at the unsampled locations using a basic radio propagation model.

**Metrics.** The RSSI error is the absolute difference between the predicted and the collected RSSI values at all test locations.

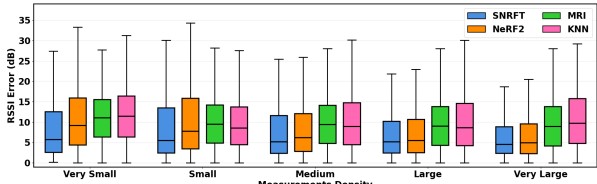

*Figure 2.* BLE RSSI Error (Lower values indicate better performance)

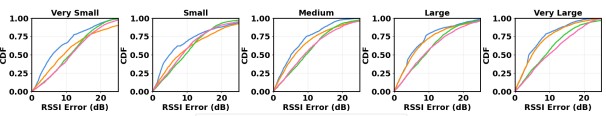

*Figure 3.* CDF of BLE RSSI Error

### 5.2.2. RESULTS

To assess performance under realistic conditions, we evaluate all methods across multiple training data densities: very small (0.175 samples/sq ft), small (0.35 samples/sq ft), medium (0.70 samples/sq ft), large ($\sim$1.50 samples/sq ft), and very large ($\sim$3.00 samples/sq ft).

Figures 2 and 3 summarize the comparative results under different wireless protocols. On the very small BLE dataset, SNRFT reduces the median RSSI error to 5.6 dB, which is approximately at least 4 dB lower than all other baselines. As the amount of training data increases, the performance gap between ray-tracing-based models (e.g., SNRFT) and ray marching-based methods (e.g., NeRF[2]) narrows. With limited data, physics-informed models benefit from strong inductive biases that constrain the solution space, yielding more accurate predictions. As datasets grow, ray-marching-based methods acquire sufficient information to learn approximate physical behaviors directly from the data. Their high expressive capacity enables them to capture complex mappings that were previously infeasible with tiny datasets.

These results indicate that SNRFT is more effective at learning and generalizing from limited data. This advantage stems from its strong physical constraints, which guide the model toward physically plausible solutions even when measurements are sparse.

## 5.3. Millimeter-Wave (mmWave) Channel Estimation

### 5.3.1. EXPERIMENTAL SETTINGS

**Task.** This task indicates SNRFT's capability in different radio-frequency bands. Given a TX sending RF signals at location $(x_{\text{TX}}, y_{\text{TX}}, z_{\text{TX}})$, the goal is to synthesize the RF signal received by the RX.

**Dataset.** We use a real-world mmWave dataset provided by RFCanvas (Chen et al., 2024), which was collected in four

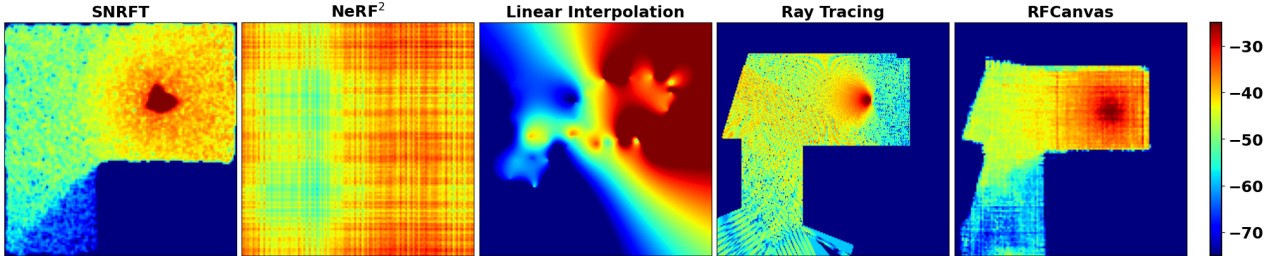

*Figure 4.* mmWave channel estimation of one indoor environment (trained with an RF sample density of $0.23\,\mathrm{sp/ft^2}$).

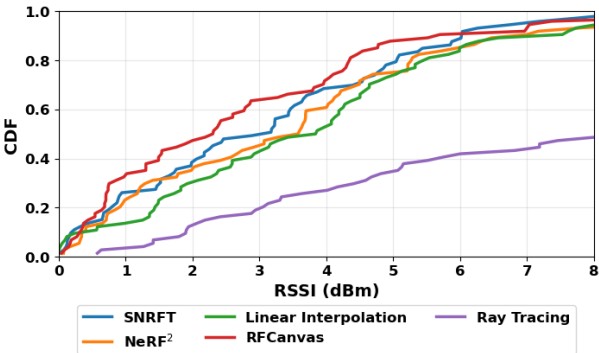

*Figure 5.* CDF curve of mmWave reconstruction Error.

real-world indoor environments. Each scene contains approximately 0.48 samples per square foot in the 60 GHz frequency band, acquired using 802.11ad-compliant MikroTik wAP 60Gx3 routers (Blanco et al., 2022).

**Baselines.** We compare SNRFT with **NeRF**$^2$ (Zhao et al., 2023), as it outperforming other approaches such as DC-GAN (Radford et al., 2015) and VAE (Kingma & Welling, 2013). Additionally, we employ **linear interpolation** from the SciPy package as a pure signal processing baseline, and **Ray tracing** (Ma et al., 2024) given the 3D mesh of the scene. **RFcanvas** (Chen et al., 2024) is included as vision-assisted baseline that fuses the vision prior of the scene.

**Metrics.** We employ the CDF of the reconstruction error, defined as the absolute difference between the predicted and collected measurements across all test locations, and provide qualitative visualizations via received-signal heatmaps.

### 5.3.2. RESULTS

Figure 4 shows the received signal heatmaps generated by different methods. Ray tracing accurately captures the overall channel structure but exhibits a median error of 7 dB even after surface material refinement, highlighting its limited ability to model complex internal propagation effects. NeRF$^2$ struggles to recover the channel structure due to its lack of explicit modeling of free-space and object boundaries. Linear interpolation better preserves structural fea-

tures than NeRF$^2$ but fails in regions affected by occlusion, multipath propagation, or sparse training data. In contrast, SNRFT reliably captures the channel structure and achieves high numerical accuracy *without* relying on any visual priors.

Figure 5 reports the CDFs of reconstruction errors across all methods. SNRFT consistently outperforms all baselines except RFcanvas (which leverages a visual prior), demonstrating its effectiveness in modeling indoor RF channels from few-shot measurements.

### 5.4. 5G Channel State Information Estimation

#### 5.4.1. EXPERIMENTAL SETTINGS

**Task.** This task demonstrates SNRFT 's effectiveness in estimating complex-valued channel state information (CSI) and antenna array setup. Given uplink complex-valued CSI, the objective is to predict the downlink CSI in 5G Orthogonal Frequency Division-Multiplexing (OFDM) modulation (Prasad, 2004). The rationale for this task lies in the assumption that both link channels are created by the same underlying physical environment (Vasisht et al., 2016).

**Dataset.** We employ a real-world open source Argos dataset (Shepard et al., 2016). The dataset is collected in outdoor environments, where a base station with 104 antennas measures CSI from eight mobile users. Each CSI measurement includes 52 subcarriers, with the first half of the 52 subcarriers (i.e., 26 subcarriers) for the uplink channel and the remained half the downlink channel (Zhao et al., 2023; Yang et al., 2025). The dataset contains 100k measurements, split into 80% for training and 20% for testing.

**Baselines.** We choose **NeRF**$^2$ (Zhao et al., 2023), **GSRF** (Yang et al., 2025), **FIRE** (Liu et al., 2021) for comparison.

**Metrics.** We adopt the Signal-to-Noise Ratio(SNR) to quantify estimated CSI quality: $\mathrm{SNR} = -10\log_{10}(\|\hat{h} - h\|_2^2/\|h\|_2^2)$, where $h$ and $\hat{h}$ are the ground truth and predicted CSI vectors, respectively. A higher positive SNR is obtained when the predicted channel gets closer to the ground truth.

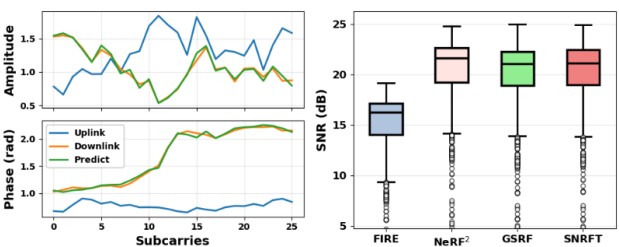

*Figure 6.* left: Channel amplitude and phase; right: Channel CSI prediction SNR.

### 5.4.2. RESULTS

For fairness, the results of the three baseline methods are taken from (Zhao et al., 2023; Yang et al., 2025). To demonstrate SNRFT's efficiency, it is trained on only 12.5% of the raw training data, and all methods are evaluated on the same test set. Figure 6 shows a representative prediction from SNRFT, where the two curves nearly overlap, indicating high prediction accuracy. The figure also compares the SNR of all four methods. SNRFT outperforms GSRF and FIRE while achieving comparable CSI synthesis quality to NeRF$^2$ using $6\times$ less training data, highlighting its superior training efficiency.

### 5.5. Ablation Study

To evaluate the effectiveness of our attenuation modeling design, we conduct an ablation study on the BLE dataset introduced in Section 5.2. Specifically, we compare the proposed physics-guided attenuation formulation in (14) against an alternative design in which the attenuation coefficient is directly regressed by a neural network without the structured formulation derived from RF propagation principles. Results under different measurement densities are summarized in Table 1.

*Table 1.* BLE RSSI error across different data density (in dB, lower is better).

| Data Density (samples/sq ft) | SNRFT | Ablation |
|:---:|:---:|:---:|
| 0.175 | **5.7569** | 7.3238 |
| 0.35 | **5.5088** | 6.4039 |
| 0.70 | **5.2135** | 6.4167 |
| 1.50 | **5.1996** | 6.7051 |
| 3.00 | **4.5711** | 5.7537 |

As shown in Table 1, the proposed SNRFT consistently achieves lower RSSI prediction error across all data densities. The improvement is particularly pronounced in sparse-measurement regimes, where the ablated model suffers from noticeably degraded generalization performance. These results suggest that incorporating physically informed at-

tenuation modeling provides a strong inductive bias that regularizes learning and improves sample efficiency.

## 6. Conclusions and Future Work

We have presented SNRFT, a novel framework that seamlessly integrates neural object representations with ray tracing for efficient and accurate RF channel modeling. By modeling RF propagation as a stochastic transport process, SNRFT captures complex RF-object interactions while inherently satisfying physical constraints such as reciprocity. As an advancement over physics-based ray tracing simulators widely adopted in the wireless community, SNRFT holds strong potential as a versatile tool for applications such as network planning, indoor localization, and channel modeling.

Despite its effectiveness, our work also has limitation that open promising avenues for future research. While we adopted exponential transport as a convenient and widely used approximation, exploring alternative non-exponential models, such as the first-passage times of Gaussian processes, represents an important direction for our further investigation.

## Acknowledgement

We thank the reviewers for their insightful comments. This work is generously supported by the UC San Diego Center for Wireless Communications, DOE DE-SC0022165, NSF 2124929, NSF 2346550, NSF 2312715, NSF 2403124.

## Impact Statement

This paper presents work whose goal is to advance the field of Machine Learning. There are many potential societal consequences of our work, none of which we feel must be specifically highlighted here.

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

# A. Ray Tracing and RF Propagation Primer

## A.1. Channel Models

Channel models are typically constructed either statistically, by defining distributions over channel attributes, or deterministically, using ray tracing. Statistical models fall short for applications involving positioning, sensing, and communications. Inspired by analogous techniques in computer graphics (Glassner, 1989), traditional ray tracing approaches (McKown & Hamilton, 1991) approximate electromagnetic wave propagation by modeling the interactions of individual rays with objects along their paths. These interactions may include reflection, diffraction, and penetration. While more efficient than directly solving Maxwell's equations, ray tracing still requires detailed environmental knowledge and is often slow for prototyping. To remain computationally tractable, these methods typically rely on hard-coded, simplified physical models, such as the knife-edge approximation for diffraction (Lee, 1997). However, such abstractions often suffer from mismatches, necessitating laborious fine-tuning and calibration with real-world data. Moreover, improving their fidelity while keeping simulations efficient is challenging. Finally, because they are inherently non-differentiable, these models cannot be integrated into closed-loop design pipelines. To address these limitations, we propose a neural model for physics-informed wireless ray tracing in this paper.

## A.2. Ray Attributes

We denote the $k$-th ray (among $K$ rays) at the $i$-th rendering segment as $\boldsymbol{r}_k^{(i)}$. For notational simplicity, we omit sub- and superscripts in the remainder of this section. The wireless ray is characterized analogously to an optical ray (e.g., by its geometric direction). In addition to wireless channel attributes, we introduce meta-level attributes that facilitate the propagation and rendering of the eventual ray received at the receiver coordinate $\boldsymbol{x}_{\mathrm{RX}}$. The complete ray representation is given by

$$\boldsymbol{r} = \big(T,\ \tau,\ \varphi,\ \boldsymbol{p}_0,\ \boldsymbol{\omega},\ t_s,\ t_{\mathrm{RX}},\ \rho_{\mathrm{RX}},\ \beta_{\mathrm{act}},\ \beta_{\mathrm{RX}}\big),$$

which can be grouped into three categories:

*(a) Wireless channel attributes.* $T$, $\tau$, and $\varphi$ are the accumulated wireless attributes. As discussed in Section 4.4, these attributes are used to construct the wireless channel response.

*(b) Ray geometry.* We additionally encode the ray's geometric representation, which governs its propagation through the scene. The ray path is defined by

$$\boldsymbol{p}(t) = \boldsymbol{p}_0 + t\boldsymbol{\omega},$$

where $\boldsymbol{p}_0$ is the origin and $\boldsymbol{\omega}$ is a unit vector denoting the ray direction. We focus on two particular solutions of $t$: $t_s$, where the ray intersects a surface (represented by the implicit function $f(\cdot)$ in our case), and $t_{\mathrm{RX}}$, where the ray becomes tangential to a reception sphere centered at a receiver with radius $\rho_{\mathrm{RX}}$.

*(c) Ray state.* To enable updates across rendering segments, we maintain two binary state variables. $\beta_{\mathrm{act}}$ indicates whether the ray remains active in the next segment (inactive rays have either reached a RX or exited the scene boundary). $\beta_{\mathrm{RX}}$ flags whether the ray has impinged on a reception sphere of a predefined radius.

## A.3. Details of Ray Tracing

**Ray–Scene Interactions.** For each ray, we are interested in its first interaction with the environment (e.g., the first object it encounters or impingement on the receiver). To this end, we consider the solutions to the line equation that defines the ray geometry:

$$\boldsymbol{p}(t) = \boldsymbol{p}_0 + t\boldsymbol{\omega}.$$

We focus on two particular solutions of $t$:

*(a) Ray-Surface Interactions.* The smallest $t > 0$ such that $\boldsymbol{p}(t)$ lies on a surface. This is obtained by performing ray–surface intersections over all scene geometry and identifying the corresponding solution $t = t_s$, which yields the updated relay location $\boldsymbol{p}_0 + t_s\boldsymbol{\omega}$.

*(b) Ray-RX Interactions.* We also consider positive solutions of $t$ for which the ray intersects the receiver, modeled as a sphere of radius $\rho_{\mathrm{RX}}$. In this case, $t$ is computed as the projection of $\boldsymbol{x}_{\mathrm{RX}}$ onto the ray trajectory $\boldsymbol{p}(t)$.

At the end of each ray–environment query, we thus obtain analytical estimates of the ray's first intersection with both the scene and, if applicable, the receiver.

**Ray–Surface Interactions.** If the ray $r^{(i)}$ (originating at $x^{(i)}$ and propagating along direction $\omega^{(i)}$) encounters an object at $x^{(i+1)}$ (as determined in the previous step), our objective is to characterize the outgoing ray that emanates from $x^{(i+1)}$. Specifically, we seek to estimate the new direction $d^{(i+1)}$ (e.g., whether the ray undergoes penetration or reflection) together with the associated change in gain (e.g., power attenuation or phase shift). This is a challenging problem, as it typically requires detailed knowledge of the surface properties (e.g., material composition) as well as their electromagnetic (EM) characteristics (e.g., frequency-dependent responses).

Building off subsection 4.3, we primarily expect specular reflections, albeit some flexibility in the signals which return to the receiver. Even though a perfectly specular signal will only return to the receiver if the reflecting surface is normal to the transmitted chirp, in actuality, there is a margin in which the signal is not perfectly specular, yet some signal still returns. Similar to commons ways of modeling diffused reflections, we follow a shifted Lambertian model (Ren et al., 2020), sometimes also called the directive model. We represent the incident direction by the unit vector $\omega_i$, the surface normal by $n$, and the ideal specular reflection direction by:

$$\omega_o = \omega_i - 2\langle \omega_i, n \rangle n. \tag{24}$$

The ray which returns to the receiver antenna is denoted as $\omega_r$, and is calculated by taking the unit vector that goes from the surface point to the receiver location.

**Termination Check.** For certain special cases, ray tracing is terminated for a subset of rays to avoid erroneous re-reception of the same ray in subsequent iterations. Namely, when ray $k$ impinges on a RX sphere of radius under $\rho_{\mathrm{RX}}$ meters. For computational efficiency, we also terminate ray tracing when a ray propagates outside the region of interest (e.g., leaving the bounded environment).

## B. Neural Scene and Material Representation

In this section, we describe the neural representation used to parameterize the scene geometry, material-dependent attenuation, phase response, and ray-launch distribution. We also summarize the main architectural choices and implementation configurations used in our experiments.

### B.1. Scene Representation

To improve computational efficiency, we consider the special case where the mean implicit function $f(x)$ is an SDF (signed distance function) (Osher et al., 2004; Gilbarg et al., 1998), which permits sphere tracing (Hart, 1996). We represent a scene implicitly via an SDF, a standard construction in level-set methods. Let $\Omega \subset X$ be a subset of a metric space $(X, d)$, and let $\partial\Omega$ denote its boundary. The point-to-set distance from $x \in X$ to $\partial\Omega$ is defined by

$$d(x, \partial\Omega) := \inf_{y \in \partial\Omega} d(x, y).$$

The corresponding signed distance function (SDF) to $\Omega$ is then

$$f(x) := \begin{cases} d(x, \partial\Omega), & x \in \Omega, \\ -d(x, \partial\Omega), & x \notin \Omega, \\ 0, & x \in \partial\Omega. \end{cases}$$

If $\Omega \subset \mathbb{R}^n$ has a piecewise smooth boundary, then $f$ is differentiable almost everywhere and satisfies the eikonal equation

$$\|\nabla f(x)\|_2 = 1.$$

### B.2. Implicit Structure Network

The scene is represented by a coordinate MLP. The structure network maps a 3D coordinate to a mean implicit field and a local feature vector,

$$G_{\Theta_g} : x \mapsto \big(f(x), h(x)\big), \tag{25}$$

where $f(\boldsymbol{x})$ is used as an SDF-like scalar field and $\boldsymbol{h}(\boldsymbol{x})$ is a $128-$dimensional vector carrying local geometric features for the material network. The implementation uses an 8-layer MLP with hidden dimension 64 and a skip connection at layer 4. Input coordinates are encoded with sinusoidal positional features

$$\gamma(\boldsymbol{x}) = \big[\sin(2^0\pi\boldsymbol{x}), \cos(2^0\pi\boldsymbol{x}), \ldots, \sin(2^{L-1}\pi\boldsymbol{x}), \cos(2^{L-1}\pi\boldsymbol{x})\big] \tag{26}$$

with $L = 6$ by default. The field is geometrically initialized either from a sphere or from the hallway prior used in the experiments.

### B.3. Material, Phase, and Ray-Launch Weights

The material network receives the same spatial coordinate and the local feature vector from the structure network:

$$M_{\boldsymbol{\Theta}_m} : (\boldsymbol{x}, \boldsymbol{h}(\boldsymbol{x})) \mapsto \big(m(\boldsymbol{x}), \varphi(\boldsymbol{x})\big), \tag{27}$$

implemented as an 8-layer MLP with hidden dimension 128 and a skip connection at layer 4. The output activations enforce reasonable ranges: $m(\boldsymbol{x}) \geq 0$ through a ReLU, and $\varphi(\boldsymbol{x}) \in [0, 2\pi]$ through a shifted tanh. The current renderer uses $m(\boldsymbol{x})$ and $\varphi(\boldsymbol{x})$ in the complex attenuation coefficient.

The renderer also learns a ray-launch weighting network

$$W_{\boldsymbol{\Theta}_w} : (\boldsymbol{\omega}, \boldsymbol{x}_{\text{TX}}) \mapsto w, \tag{28}$$

implemented as a 4-layer MLP with hidden dimension 64 and positional encodings for both direction and transmitter location. Its positive outputs are normalized across the sampled ray set so that emitted rays form a learned angular importance distribution around each transmitter.

## C. Differentiable RF Renderer

### C.1. Ray Emission

For each transmitter position $\boldsymbol{x}_{\text{TX}}$, the renderer partitions azimuth and elevation into a regular grid and samples multiple rays per angular cell. In the default configuration, it uses 72 azimuth bins, 36 elevation bins, and 10 samples per angular cell. Each sampled point lies on a small sphere of radius $5 \times 10^{-3}$ around the transmitter, and the corresponding ray direction is the normalized vector from $\boldsymbol{x}_{\text{TX}}$ to that sampled point. The learned weight network assigns the normalized launch weight for each ray.

### C.2. Ray-Surface Interactions

With an SDF representation, we can efficiently apply sphere tracing to compute ray–surface intersection points from the ray origins and directions. Specifically, we seek the smallest $t > 0$ such that $|f(\boldsymbol{p}(t))| < \epsilon$. The corresponding pseudo-code is provided in Algorithm 1.

### C.3. Ray-Rx Interactions

For each ray bounce/segment, we additionally account for the case where the ray reaches the receiver by checking for positive values of $t$ at which the ray intersects the receiver region. We model the receiver as a sphere of radius $\rho_{\text{RX}}$ centered at $\boldsymbol{x}_{\text{RX}}$. For computational efficiency, we terminate rays that hit the receiver during sphere tracing, rather than performing a separate ray–receiver intersection test after tracing. The corresponding pseudo-code is provided in Algorithm 2.

### C.4. Batched Path Identification

Rather than relying on hand-crafted rules for penetration and reflection, we learn the corresponding RF interaction effects through spatially varying material and EM attributes. Concretely, the neural representation maps each spatial coordinate to a material coefficient $m(\cdot)$ and a phase rotation $\varphi(\cdot)$. Together with the attenuation representation in (14) and (16), these quantities determine the gain modification induced at surface interactions.

After a ray reaches a surface, the renderer propagates either a reflected ray or a penetrative ray. The reflected direction is computed by (24). Reflection and penetration are then selected by Monte Carlo sampling (Chen et al., 2025). Specifically,

**Algorithm 1** Sphere Tracing

1: **Input:** Ray origins $\{o_i\}_{i=1}^B$, ray directions $\{d_i\}_{i=1}^B$, where $B$ is the number of the rays.
2: **Input:** SDF of the scene $f(\cdot)$, region of interest $\Omega$.
3: **Input:** Intersection threshold $\epsilon$, max steps $J$, coefficient $\beta < 1$.
4: **Output:** Intersection points $\{q_i\}_{i=1}^B$.
5: **for** $i = 1$ to $B$ **do**
6:   **for** $j = 1$ to $J$ **do**
7:     **if** $o_i \in \Omega$ **then**
8:       **if** $|f(o_i)| < \epsilon$ **then**
9:         $q_i \leftarrow o_i$
10:         **break**
11:       **else**
12:         $o_i \leftarrow o_i + \beta|f(o_i)|d_i$
13:       **end if**
14:     **else**
15:       $q_i \leftarrow$ NaN
16:       **break**
17:     **end if**
18:   **end for**
19: **end for**

**Algorithm 2** Compute Ray-Receiver Intersection Mask

1: **Input:** Intermediate points $\{o_i\}_{i=1}^B$, ray directions $\{d_i\}_{i=1}^B$, where $B$ is the number of the rays.
2: **Input:** SDF of the scene $f(\cdot)$, receiver center $x_{\text{RX}}$, receiver radius $\rho_{\text{RX}}$.
3: **Output:** Boolean mask $\{\text{mask}_i\}_{i=1}^B$.
4: **for** $i = 1$ to $B$ **do**
5:   $t_i \leftarrow (x_{\text{RX}} - o_i)^\top d_i$
6:   $v_i \leftarrow \|(o_i + t_i d_i) - x_{\text{RX}}\|_2$
7:   $\text{mask}_i \leftarrow (t_i \leq f(o_i)) \wedge (v_i \leq \rho_{\text{RX}})$
8: **end for**

we define

$$c_{\text{ref}} = |\langle \omega_{\text{ref}}, n \rangle|, \quad c_{\text{pen}} = |\langle \omega, n \rangle|, \tag{29}$$

and sample a penetrative continuation with probability

$$p_{\text{pen}} = \frac{c_{\text{ref}}}{c_{\text{ref}} + c_{\text{pen}}}. \tag{30}$$

Otherwise, the ray is reflected. Reflected rays are offset by $+2\epsilon n$ and propagated along $\omega_{\text{ref}}$, while penetrative rays are offset by $-2\epsilon n$ and continue along the incident direction $\omega$. These small offsets prevent immediate self-intersections with the same boundary.

### C.5. Implementation Details & Configurations

*Table 2.* Configuration used in the implementation.

| Configuration | Value | Description |
|---|---|---|
| X_RANGE | $(0,1)$ | Normalization range for the $x$ coordinate. |
| Y_RANGE | $(0,1)$ | Normalization range for the $y$ coordinate. |
| Z_RANGE | $(0,1)$ | Normalization range for the $z$ coordinate. |

| Configuration | Value | Description |
|---|---|---|
| MAX_ITERS | 128 | Maximum number of sphere-tracing iterations. |
| MAX_DISTANCE | 2.0 | Maximum ray propagation distance. |
| MAX_STEP | 0.02 | Maximum step size during sphere tracing. |
| EPSILON | $2 \times 10^{-2}$ | Surface hit threshold. |
| SPACE_BOUNDARIES | $[1.0, 1.0, 1.0]$ | Normalized scene boundary size. |
| SCALING | 1.0 | Scaling coefficient used in attenuation computation. |
| NUM_RAYS_AZIMUTH | 72 | Number of azimuth bins for ray emission. |
| NUM_RAYS_ELEVATION | 36 | Number of elevation bins for ray emission. |
| RAYS_PER_AREA | 10 | Number of rays sampled per angular bin. |
| RADIUS | $5 \times 10^{-3}$ | Initial emission sphere radius around the transmitter. |
| MAX_BOUNCES | 3 | Maximum number of ray interactions. |
| SAMPLE_SPACING | $2 \times 10^{-3}$ | Sampling interval for penetration paths. |
| RX_RADIUS | 0.02 | Receiver sphere radius. |
| TX_SIGNALS | 1.0 | Transmitted signal amplitude. |

# D. Proof of Theorem 4.4

*Proof.* We begin with the key intermediate results and the corresponding notations in the proof of Theorem 4 in (Miller et al., 2024). Please refer to (Miller et al., 2024) Supplementary Material section F.1 for a more detailed illustration.

The object-inside propagation distribution $p_{\boldsymbol{x},\boldsymbol{\omega}}(t)$ is the probability density that, starting from $I(\boldsymbol{x}) = 1$, the first $1 \to 0$ transition of the indicator function along the ray (that is, the first intersection) occurs at the distance $t$. For this distance to be an exponential random variable, $I_{\boldsymbol{x},\boldsymbol{\omega}}(t)$ must be a continuous-time discrete-space Markov process. The object-inside propagation distance $d^*_{\boldsymbol{x},\boldsymbol{\omega}}$ is an exponential random variable, which equals the first jump time $t^*_{\boldsymbol{x},\boldsymbol{\omega}} \triangleq \min\{t \in [0,\infty) : I_{\boldsymbol{x},\boldsymbol{\omega}}(t) = 0 | I_{\boldsymbol{x},\boldsymbol{\omega}}(t) = 1\}$. Let $\sigma_{10}(\boldsymbol{x},\boldsymbol{\omega})$ be the object-inside transition rate of the exponential distribution of the first jump time $t^*_{\boldsymbol{x},\boldsymbol{\omega}}$. Analogously, $\sigma_{01}(\boldsymbol{x},\boldsymbol{\omega})$ denotes the transition rate of the exponential distribution of the first jump time for the $0 \to 1$ transition. Following (Miller et al., 2024), solving *Kolmogorov equations*, we have:

$$-\boldsymbol{\omega} \cdot \nabla o(\boldsymbol{x}) = -(1 - o(\boldsymbol{x}))\sigma_{01}(\boldsymbol{x},\boldsymbol{\omega}) + o(\boldsymbol{x})\sigma_{10}(\boldsymbol{x},\boldsymbol{\omega}), \tag{31}$$

$$\boldsymbol{\omega} \cdot \nabla o(\boldsymbol{x}) = -(1 - o(\boldsymbol{x}))\sigma_{01}(\boldsymbol{x},-\boldsymbol{\omega}) + o(\boldsymbol{x})\sigma_{10}(\boldsymbol{x},-\boldsymbol{\omega}). \tag{32}$$

**Reciprocity.** Under the basic reciprocity assumption inside object, we have $T_{\boldsymbol{x},\boldsymbol{\omega}}(t) = T_{\boldsymbol{y},-\boldsymbol{\omega}}(t)$ if $y = \boldsymbol{r}_{\boldsymbol{x},\boldsymbol{\omega}}(t)$ with $I(\boldsymbol{x}) = 1$. Then for any $\boldsymbol{\omega}$, Eq. 8 implies:

$$\int_0^t \sigma_{10}(\boldsymbol{r}_{\boldsymbol{x},\boldsymbol{\omega}}(s),\boldsymbol{\omega})ds = \int_0^t \sigma_{10}(\boldsymbol{r}_{\boldsymbol{y},-\boldsymbol{\omega}}(s),-\boldsymbol{\omega})ds. \tag{33}$$

Differentiating with respect to the distance and using the Leibniz integral rule, we have

$$\sigma_{10}(\boldsymbol{r}_{\boldsymbol{x},\boldsymbol{\omega}}(t),\boldsymbol{\omega}) = \sigma_{10}(\boldsymbol{r}_{\boldsymbol{y},-\boldsymbol{\omega}}(t),-\boldsymbol{\omega}) + \int_0^t \frac{d}{dt}\sigma_{10}(\boldsymbol{r}_{\boldsymbol{y},-\boldsymbol{\omega}}(s),-\boldsymbol{\omega})ds. \tag{34}$$

Note that

$$\frac{d}{dt}\sigma_{10}(\boldsymbol{r}_{\boldsymbol{y},-\boldsymbol{\omega}}(s),-\boldsymbol{\omega}) = -\boldsymbol{\omega} \cdot \frac{\partial}{\partial \boldsymbol{z}}\sigma_{10}(\boldsymbol{z},-\boldsymbol{\omega})\big|_{\boldsymbol{z}=\boldsymbol{r}_{\boldsymbol{y},-\boldsymbol{\omega}}(s)} \leq B, \tag{35}$$

where $B \in \mathbb{R}$ is a certain upper bound. Since $\sigma_{10}(\boldsymbol{x},\boldsymbol{\omega})$ is differentiable with respect to the point $\boldsymbol{x}$. Taking $t = 0$, we have

$$\sigma_{10}(\boldsymbol{x},\boldsymbol{\omega}) = \sigma_{10}(\boldsymbol{x},-\boldsymbol{\omega}). \tag{36}$$

Then Eq. 31, 32 and 36 form an underdetermined linear system that admits the solution

$$\begin{bmatrix} \sigma_{01}(\boldsymbol{x},\boldsymbol{\omega}) \\ \sigma_{01}(\boldsymbol{x},-\boldsymbol{\omega}) \\ \sigma_{10}(\boldsymbol{x},\boldsymbol{\omega}) \\ \sigma_{10}(\boldsymbol{x},-\boldsymbol{\omega}) \end{bmatrix} = \begin{bmatrix} \tau(\boldsymbol{x},\boldsymbol{\omega}) + \mathbf{1}\{\boldsymbol{\omega} \cdot \nabla o(\boldsymbol{x}) \geq 0\} \cdot \frac{2|\boldsymbol{\omega} \cdot \nabla o(\boldsymbol{x})|}{1-o(\boldsymbol{x})} \\ \tau(\boldsymbol{x},\boldsymbol{\omega}) + \mathbf{1}\{\boldsymbol{\omega} \cdot \nabla o(\boldsymbol{x}) < 0\} \cdot \frac{2|\boldsymbol{\omega} \cdot \nabla o(\boldsymbol{x})|}{1-o(\boldsymbol{x})} \\ \frac{\tau(\boldsymbol{x},\boldsymbol{\omega})(1-o(\boldsymbol{x}))+|\boldsymbol{\omega} \cdot \nabla o(\boldsymbol{x})|}{o(\boldsymbol{x})} \\ \frac{\tau(\boldsymbol{x},\boldsymbol{\omega})(1-o(\boldsymbol{x}))+|\boldsymbol{\omega} \cdot \nabla o(\boldsymbol{x})|}{o(\boldsymbol{x})} \end{bmatrix}, \tag{37}$$

where $\mathbf{1}\{\cdot\}$ denotes the indicator function, and $\tau(\boldsymbol{x}, \boldsymbol{\omega}) \geq 0$ is a free variable.

**Reversibility.** The above equation contains the undetermined function $\tau(\boldsymbol{x}, \boldsymbol{\omega})$, which is not fixed by enforcing reciprocity and exponential transport alone. To identify $\tau(\boldsymbol{x}, \boldsymbol{\omega})$, observe that for any two points $\boldsymbol{x}, \boldsymbol{y}$, Eq. 32 permits computing the occupancy $o$ at one point from that at the other by integrating along the straight line segment joining them. Since these occupancies represent probabilities of the same underlying random field $I$, the result must be consistent regardless of integration direction—whether from $\boldsymbol{x}$ to $\boldsymbol{y}$ or from $\boldsymbol{y}$ to $\boldsymbol{x}$. In ray notation, this bidirectional consistency implies the following constraint: Given $\boldsymbol{x}, \boldsymbol{\omega}$, we define $O_{\boldsymbol{x}, \boldsymbol{\omega}}(t) \triangleq o(\boldsymbol{x} + \boldsymbol{\omega} t)$ for any $t \in \mathbb{R}$,

- In one direction, we start at $\boldsymbol{r}_{\boldsymbol{x}, \boldsymbol{\omega}}(0) = \boldsymbol{r}_{\boldsymbol{x}, \boldsymbol{x} \to \boldsymbol{y}}(0) = \boldsymbol{x}$ with initial condition $O_{\boldsymbol{x}, \boldsymbol{x} \to \boldsymbol{y}}(0) = o(\boldsymbol{x})$. Note that $\frac{d}{dt} O_{\boldsymbol{x}, \boldsymbol{x} \to \boldsymbol{y}}(t) = \boldsymbol{\omega} \cdot \nabla_{\boldsymbol{z}} o(\boldsymbol{z})\big|_{\boldsymbol{z} = \boldsymbol{x} + \boldsymbol{\omega} t}$. Integrating the left hand side of Eq. 32 with respect to $t$ along the ray from point $\boldsymbol{x}$ to $\boldsymbol{r}_{\boldsymbol{x}, \boldsymbol{x} \to \boldsymbol{y}}(\|\boldsymbol{y} - \boldsymbol{x}\|) = \boldsymbol{y}$, the integration results should be $O_{\boldsymbol{x}, \boldsymbol{x} \to \boldsymbol{y}}(\|\boldsymbol{y} - \boldsymbol{x}\|) = o(\boldsymbol{y})$.

- Analogously, in the reverse direction, we start at $\boldsymbol{r}_{\boldsymbol{y}, -\boldsymbol{\omega}}(0) = \boldsymbol{y}$ with initial condition $O_{\boldsymbol{y}, \boldsymbol{y} \to \boldsymbol{x}}(0) = o(\boldsymbol{y})$ and integrating the left hand side of Eq. 32 with respect to $t$ along the ray until we reach $\boldsymbol{r}_{\boldsymbol{y}, \boldsymbol{y} \to \boldsymbol{x}}(\|\boldsymbol{y} - \boldsymbol{x}\|) = \boldsymbol{x}$. The integration results should be $\mathcal{V}_{\boldsymbol{y}, \boldsymbol{y} \to \boldsymbol{x}}(\|\boldsymbol{y} - \boldsymbol{x}\|) = v(\boldsymbol{x})$.

Consider the homogeneous linear ordinary differential equation $\boldsymbol{\omega} \cdot \nabla o(\boldsymbol{x}) = g(\boldsymbol{x}, \boldsymbol{\omega}) \cdot o(\boldsymbol{x})$, where $g(\boldsymbol{x}, \boldsymbol{\omega}) = g(\boldsymbol{x}, -\boldsymbol{\omega})$. This ODE admits solutions of exponential form, i.e.

$$\underbrace{O_{\boldsymbol{x}, \boldsymbol{x} \to \boldsymbol{y}}(\|\boldsymbol{y} - \boldsymbol{x}\|)}_{o(\boldsymbol{y})} = \underbrace{O_{\boldsymbol{x}, \boldsymbol{x} \to \boldsymbol{y}}(0)}_{o(\boldsymbol{x})} \cdot \exp\left(\int_0^{\|\boldsymbol{y} - \boldsymbol{x}\|} g(\boldsymbol{x} + \boldsymbol{\omega} t, \boldsymbol{\omega}) dt\right), \tag{38}$$

$$\underbrace{O_{\boldsymbol{y}, \boldsymbol{y} \to \boldsymbol{x}}(\|\boldsymbol{y} - \boldsymbol{x}\|)}_{o(\boldsymbol{x})} = \underbrace{O_{\boldsymbol{y}, \boldsymbol{y} \to \boldsymbol{x}}(0)}_{o(\boldsymbol{y})} \cdot \exp\left(-\int_0^{\|\boldsymbol{y} - \boldsymbol{x}\|} g(\boldsymbol{y} - \boldsymbol{\omega} t, -\boldsymbol{\omega}) dt\right), \tag{39}$$

which satisfies the forward/backward integration for reversibility guarantee. In this case, the right hand side of Eq. 32 should be proportional to $o(\boldsymbol{x})$, i.e., we have $\tau(\boldsymbol{x}, \boldsymbol{\omega}) \propto \frac{o(\boldsymbol{x})}{1 - o(\boldsymbol{x})}$.

**Material/Medium Dependency.** Moreover, we need to take material property into account to distinguish different attenuation magnitude degradation in different material. However, a practical object's material properties are distributed unevenly across its interior and are often not measurable. Therefore, we set $\tau(\boldsymbol{x}, \boldsymbol{\omega}) = \frac{m(\boldsymbol{x}) o(\boldsymbol{x}) |\boldsymbol{\omega} \cdot \nabla o(\boldsymbol{x})|}{1 - o(\boldsymbol{x})}$, where $m(\boldsymbol{x})$ is a coordinate-dependent coefficient that accounts for material/medium property. In this way, we have

$$\sigma_{10}(\boldsymbol{x}, \boldsymbol{\omega}) = \frac{(m(\boldsymbol{x}) o(\boldsymbol{x}) + 1) |\boldsymbol{\omega} \cdot \nabla o(\boldsymbol{x})|}{o(\boldsymbol{x})}. \tag{40}$$

Equivalently, using $o(\boldsymbol{x}) = 1 - v(\boldsymbol{x})$, we can write

$$\sigma_{10}(\boldsymbol{x}, \boldsymbol{\omega}) = \frac{(m(\boldsymbol{x})(1 - v(\boldsymbol{x})) + 1) |\boldsymbol{\omega} \cdot \nabla v(\boldsymbol{x})|}{1 - v(\boldsymbol{x})} \tag{41}$$

This concludes the proof. □

