# OpenReview forum: "Stochastic Neural Ray Tracing for Radio Frequency Channel Modeling"
_ICML.cc/2026/Conference — ICML 2026 regular_

### Official Review · Reviewer_VFE3 · 2026-03-06

**Soundness:** 3
**Presentation:** 3
**Significance:** 3
**Originality:** 3
**Overall Recommendation:** 5
**Confidence:** 3

**Summary:**

The authors propose a learning-based method for simulating radio frequency propagation. It relies on ray tracing in which the signal attenuation along the ray path is derived from the stochastic object framework.
A neural field is leveraged to estimate the geometry of the scene through a SDF. Inside an object, the neural field also estimates the atenuation parameters and phase shift at any location along the ray path.

**Compliance With Llm Reviewing Policy:**

Affirmed.

**Final Justification:**

The authors provided missing ablation studies, which addressed my main concern regarding this article. The other weaknesses and questions I had have also been addressed in the rebuttal. In light of the approach's originality, its solid theoretical underpinnings, and the experiments conducted on various benchmarks, I raise my overall recommendation to 5.

**Key Questions For Authors:**

1. To address weakness 1, could the authors conduct an ablation study on one of the benchmarks they focus on? Specifically, could the authors provide the results of their network trained to predict the whole attenuation $\sigma$ instead of $m$, as in RFScape, and the results of a naive neural field approach trained with the same architecture and hyperparameters as SNRFT?
2. Could the authors correct their analysis of the results in Figure 6 as suggested in weakness 2? Regarding experiment 5.4, could they also provide results of SNRFT trained on more 12.5% of the training set to confirm that GSRF, NERF² and SNRFT with 12.5% of the training set not have already saturated the benchmark?
3. Could the authors confirm whether all the qualitative results in Figure 4 have the same x and y axes and update Figure 4 as suggested in weakness 3?
4. Do the authors plan to release their code?

**Limitations:**

yes

**Strengths And Weaknesses:**

# Strengths:
1. The authors leverage a rigorous theoretical framework to provide a strong inductive bias to a novel neural field-based approach for simulating RF propagation.
2. Experiments with diverse real-world datasets highlight the improved performance of SNRFT compared to existing methods, whether learning-based or not, and demonstrate its superior sample efficiency.
3. Overall, the paper is clear, well-presented, well-illustrated, and well-written.

# Weaknesses:
1. The authors have not conducted an ablation study to rigorously justify, with experiments, their approach of object attenuation relative to a simpler back-box method which uses the same architecture and hyperparameters as SNRFT. Providing such an ablation would ensure the performance gain cannot be explained by network architecture or hyperparameter tuning.
2. Analysis of Figure 6 (right) is misleading: according to the box plot, SNRFT with 12.5% of the training set *only* outperforms FIRE while achieving comparable results as NERF² *and* GSRF. SNRFT with 12.5% of the training set is marginally above GSRF, but it is not significant; otherwise, the authors should have also said that SNRFT with 12.5% of the training set is worse than NERF². Regarding Figure 6, it would also have been interesting to provide the results of SNRFT trained with 100% of the training set in order to assess the margin relative to the other approaches.
3. The qualitative results in Figure 4 could have been better presented. I recommend that the authors provide the unit of the color scale as well as the x and y axes for each image. Otherwise, it appears that the image corresponding to SNRFT has been cropped and resized.

# Minor weaknesses:
1. The stopping condition of the ray tracing algorithm, $\epsilon$ and the maximum iteration/depth mentioned in section 4.4, are not given in the implementation details.
2. In theorem 4.4, the relation between $I$, $I_{xw}$, and $I_{xw}(t)$ is not clearly defined.

---

> ### Author Rebuttal · Authors · 2026-03-31
>
> We sincerely thank the reviewer VFE3 for the constructive comments and suggestions. We provide additional experimental comparisons and explanations below to address your concerns.
>
> **For Question 1:**
>
> To investigate the effect of this design choice, we compare the proposed attenuation formulation (Eq.14) with an ablated variant in which the attenuation $\sigma$ is directly predicted by a neural network. The corresponding results on the BLE dataset under varying data densities are presented in the **Table 1**: <https://anonymous.4open.science/r/ICML2026_Rebuttal-05F5>.
>
> As observed, SNRFT consistently achieves lower RSSI error than the ablated model across all settings. The performance gap is particularly noticeable in the low-data regime, suggesting that the proposed formulation provides a beneficial inductive bias that improves generalization when measurements are limited.
>
> **For Question 2:**
>
> Based on our experimental results, SNRFT trained on 12.5% of the data is only slightly inferior to NeRF$^2$ trained on the full dataset. We will revise the analysis of Figure 6 accordingly. Due to the limited time during the rebuttal period, we have retrained NeRF$^2$ on 12.5% of the training set. SNRFT achieves higher CSI prediction SNR (21.33 dB) compared to NeRF$^2$ (19.85 dB). These results help provide a more direct comparison in the low-data regime. We will include the results for all methods trained on the full dataset in the final version of this paper.
>
> **For Question 3:**
>
> Yes, all the qualitative results in Figure 4 share the same x and y axes. The spatial ranges of both axes are normalized to [0,1], and the color scale is reported in dB. For RFCanvas and ray tracing, the ground-truth 3D geometry is provided, and values outside the scene boundaries are therefore masked. In contrast, our method learns an approximation of the underlying geometry, which may lead to differences near the boundaries. We will update Figure 4 according to the reviewer’s suggestions.
>
> **For Question 4:**
>
> Yes, we will release the codes.

---

> > ### Author Rebuttal · Reviewer_VFE3 · 2026-04-01
> >
> > The authors have adequately addressed my concerns, particularly with regard to the missing ablation study, which was the main weakness of the original submission. In light of the new results presented in Table 1 of the rebuttal, I will raise my score to 5. Congratulations to the authors for their great job.
> >
> > I have a minor point that could further strengthen your paper, could the authors add an ablation in which $v(x)$ in equation (11) is predicted instead of $f(x)$ in equation (14)?

---

> > > ### Author Response · Authors · 2026-04-06
> > >
> > > Thank you very much for your positive feedback and for raising your score. We truly appreciate your recognition of our efforts.
> > >
> > > We thank the reviewer for the insightful suggestion on further strengthening the paper. The proposed ablation that predicting $v(\mathbf{x})$ in Eq. (11) instead of $f(\mathbf{x})$ in Eq. (14) is indeed valuable. In response, we would like to clarify that $v(\mathbf{x})$ and $f(\mathbf{x})$ are in one-to-one correspondence under our formulation (Eq. (13)). Although $v(\mathbf{x})$ and $f(\mathbf{x})$ have equivalent expressive power, $f(\mathbf{x})$ provides a geometry-aware parameterization that enables efficient ray tracing and stable gradient-based optimization. As discussed in Appendix B.1, we adopt a special case where $f(\mathbf{x})$ is modeled as a signed distance function (SDF), which enables the use of sphere tracing instead of naive tracing for efficient intersection computations. Moreover, the first-order gradients of the SDF provide useful geometric regularization, which leads to smoother and more stable geometry representations compared to directly modeling $v(\mathbf{x})$.
> > >
> > > We will include this ablation in the revised version and report the corresponding results. We will also add a brief discussion to highlight the differences and insights gained from this comparison.
> > >
> > > Thank you again for your helpful comments and support.

---

### Official Review · Reviewer_AXJy · 2026-03-08

**Soundness:** 3
**Presentation:** 3
**Significance:** 3
**Originality:** 4
**Overall Recommendation:** 4
**Confidence:** 3

**Summary:**

This paper proposes a wireless channel modeling framework called SNRFT. It analyzes the challenge of balancing the physical interpretability of ray tracing with the high fitting capability of neural representations in complex electromagnetic environments. The authors astutely observe that existing models either become trapped in black-box neural agents lacking physical constraints, or rely excessively on manually designed interaction rules, making it difficult to handle uncertainties in the environment. SNRFT aims to establish a unified descriptive scheme based on random transmission theory, bridging these two approaches.

**Compliance With Llm Reviewing Policy:**

Affirmed.

**Final Justification:**

Thanks to the author for resolving all my concerns, I remain my evaluation unchanged.

**Key Questions For Authors:**

I have a slight concern: the current model is still limited by the exponential transport assumption, and its general applicability in very special environments such as non-exponential fading needs further verification. Could you provide some brief explanations?

**Limitations:**

yes

**Strengths And Weaknesses:**

A central concept examined in this article is modeling radio frequency (RF) propagation as a stochastic transmission process, deriving an attenuation function that satisfies reciprocity and reversibility constraints through stochastic object theory. Compared to visual rendering methods like NeRF, SNRFT's uniqueness lies in abstracting material properties into rate parameters within transmission dynamics and utilizing differentiable ray tracing pipelines to achieve co-optimization of geometry (SDF) and material electromagnetic properties. This modeling approach is not only mathematically rigorous but, more importantly, provides a statistical physics-based explanation for RF signal penetration and loss, rather than simple numerical fitting.
The experimental data is highly convincing. SNRFT demonstrates extremely high data efficiency in multiple tasks, including BLE RSSI and millimeter-wave channel estimation. Especially under small sample conditions (e.g., using only 12.5% ​​of the training data), the method still achieves state-of-the-art (SOTA) performance and achieves approximately 4 dB of error optimization in BLE experiments. This improved sample efficiency strongly demonstrates the crucial role of physically heuristic inductive bias in reducing the search space and improving generalization ability.

---

> ### Author Rebuttal · Authors · 2026-03-31
>
> We sincerely thank the reviewer AXJy for the time and effort in reviewing our paper. We appreciate your positive comments on our work.
>
> The exponential transport assumption is consistent with the physical model described in the ITU-R P.2040-1 recommendation [1], which characterizes the exponential decay of the electric field with distance in free space. Therefore, it serves as a reasonable and well-established assumption for modeling channel attenuation and has been widely adopted in prior work [2-4]. We note that the exponential transport assumption is not introduced by our work, related formulations have been discussed in Section 3. At the same time, we acknowledge that exploring alternative non-exponential models is an important direction for future research, although it is beyond the scope of the current manuscript. This limitation has been discussed in Section 6.
>
> **References:**
>
> [1] ITU-R. (2015). Effects of building materials and structures on radiowave propagation above about 100 MHz (Recommendation ITU-R P.2040-1).
>
> [2] Zhao, X., An, Z., Pan, Q., \& Yang, L. (2023, October). Nerf2: Neural radio-frequency radiance fields. In Proceedings of the 29th Annual International Conference on Mobile Computing and Networking (pp. 1-15).
>
> [3] Lu, H., Vattheuer, C., Mirzasoleiman, B., \& Abari, O. (2024). Newrf: A deep learning framework for wireless radiation field reconstruction and channel prediction. arXiv preprint arXiv:2403.03241.
>
> [4] Chen, X., Feng, Z., Sun, K., Qian, K., \& Zhang, X. (2024, November). Rfcanvas: Modeling rf channel by fusing visual priors and few-shot rf measurements. In Proceedings of the 22nd ACM Conference on Embedded Networked Sensor Systems (pp. 464-477).

---

> > ### Author Rebuttal · Reviewer_AXJy · 2026-04-04
> >
> > I thank the authors for their detailed and constructive rebuttal. I will maintain my score.

---

> > > ### Author Response · Authors · 2026-04-06
> > >
> > > We sincerely thank the reviewer for the continued engagement and for acknowledging the progress in our revision.

---

### Official Review · Reviewer_fW1g · 2026-03-11

**Soundness:** 3
**Presentation:** 2
**Significance:** 3
**Originality:** 4
**Overall Recommendation:** 5
**Confidence:** 3

**Summary:**

This paper proposes **SNRFT**, a neural framework for RF channel modeling that combines differentiable ray tracing with learned scene representations. The core idea is to model RF propagation as a **stochastic transport process**, from which attenuation is derived via a probabilistic object/material formulation, while also imposing **reciprocity-aware physical structure**. The model uses neural implicit representations for geometry, material, and phase, embedded into a ray-tracing pipeline handling reflection and penetration. Empirically, the method is evaluated on **three real-world RF tasks**: BLE RSSI prediction, indoor mmWave channel estimation, and 5G CSI estimation. The results suggest strong performance, especially in **sparse-data regimes**, and competitive or better accuracy than prior RF neural rendering baselines in several settings.

**Compliance With Llm Reviewing Policy:**

Affirmed.

**Key Questions For Authors:**

1. Can you provide targeted ablations for the **reciprocity constraint**, **stochastic attenuation model**, and **phase module** to show which theoretical ingredients are empirically necessary?
2. In the CSI experiment, were baselines reproduced under the **same setup**, or are the reported numbers drawn from prior work? If the latter, can you include at least one directly reproduced baseline?
3. Which **RF effects** are explicitly modeled in practice, and how are diffraction and diffuse scattering treated?
4. Can you provide more **systematic quantitative summaries** for the mmWave experiment rather than relying mainly on figures and CDFs?

**Limitations:**

The paper should discuss limitations more explicitly, especially the limited ablation support for its most distinctive theoretical ideas, the partially indirect CSI comparison protocol, the restricted treatment of RF effects beyond reflection/penetration, the presentation difficulty of the theory section, and the fact that evaluation is limited to **indoor scenarios**.

**Strengths And Weaknesses:**

**Strengths**

1. The paper addresses an **important and well-motivated problem**, with relevance to communications, sensing, and deployment planning. It is well positioned between hand-engineered ray tracing and black-box neural surrogates.
2. The paper is **highly original**. The stochastic-transport view, probabilistic attenuation derivation, and explicit reciprocity-aware design go beyond standard adaptations of NeRF/3DGS-style methods to RF.
3. The model design is **coherent and physically meaningful**, with a sensible split between geometry, material, and phase.
4. The empirical study is **broad**, spanning BLE, mmWave, and 5G CSI tasks, which supports the claim that the framework is not benchmark-specific.
5. There is useful evidence of **sample efficiency**, especially in the BLE setting and in the CSI experiment where the method is reported to perform well with substantially less data.

**Weaknesses**

1. The paper’s **strongest theoretical claims are not sufficiently isolated empirically**. In particular, the value of the stochastic-transport formulation and the reciprocity constraint is not demonstrated through targeted ablations.
2. The **CSI comparison is not fully controlled**, since some baseline results are taken from prior work rather than reproduced under a common setup. This weakens the strength of the efficiency claim.
3. The paper lacks **component-wise diagnostic analysis**. Missing ablations include removing reciprocity, replacing stochastic attenuation with a deterministic alternative, and removing phase prediction.
4. Some empirical evidence, especially in the mmWave section, is presented more **qualitatively than quantitatively**. More tabular summaries would make the results easier to assess.
5. The actual physical scope is **narrower than the framing suggests**: the implementation emphasizes reflection and penetration, while diffraction and diffuse scattering are less clearly handled.
6. **Presentation remains a real weakness**. The high-level idea is compelling, but the theory section is difficult to parse and the notation/writing quality could be improved substantially.

---

> ### Author Rebuttal · Authors · 2026-03-31
>
> We sincerely thank the reviewer fW1g for the time and effort in reviewing our paper. We greatly appreciate the positive feedback. We hope the following responses can resolve your questions and concerns.
>
> **For Question 1:**
>
> The stochastic transport formulation and the reciprocity/reversibility constraints are tightly integrated in our framework, jointly enabling a closed-form expression of the attenuation. In particular, stochastic transport modeling provides an underdetermined system (Appendix C, Eq.21-22), while the additional reciprocity and reversibility lead to a physically consistent solution. Consequently, isolating these components individually may not yield a meaningful representation.
>
> To examine the effect of this design, we compare the proposed attenuation formulation (Eq.14) with an ablated version where attenuation is directly predicted by a neural network. The results on the BLE dataset under different data densities are summarized in the **Table 1**: <https://anonymous.4open.science/r/ICML2026_Rebuttal-05F5>.
>
> From the results, SNRFT consistently outperforms the ablation without stochastic modeling across all data regimes, with a more pronounced advantage in the low-data setting. The performance gap for the ablation without the phase module is less significant, which is expected since RSSI prediction primarily depends on signal magnitude and does not directly supervise phase information.
>
> Unlike optical signals, the phase in a wireless channel cannot be discarded, as it encodes relative propagation information, such as time delays and path differences, that determines the coherent superposition of signals at the receiver (RX). Removing the phase would disrupt the interference structure of the channel and result in a loss of essential information. For example, in CSI experiments, the task requires predicting complex-valued downlink CSI.
>
> **For Question 2:**
>
> In the CSI experiments, two of the baselines (i.e., NeRF$^2$ and GSRF) are reproduced under the same experimental setup. The results for FIRE are taken from prior work, as its implementation is not publicly available.
>
> **For Question 3:**
>
> Due to the relatively large wavelengths of RF signals (on the order of $10^{-3}$m to $10^{-1}$m), many surfaces can be treated as electrically smooth, such that propagation is often dominated by specular reflections rather than diffuse scattering [1]. Consequently, our theoretical framework primarily models specular reflection and penetration. We emphasize that this model does take into account the multipath propagation effects where the signals bounce among reflecting surfaces.
>
> In practice, we also account for first-order diffraction and scattering effects by approximating them as reflections with perturbed directions. While this approximation allows us to partially capture their impact within the ray-tracing framework, it does not fully represent the complexity of diffraction and scattering in highly occluded environments. Extending the framework to more faithfully model these effects is an important direction for future work. We will further clarify these implementation details and associated limitations in the revised manuscript.
>
> **For Question 4:**
>
> Yes, we summarize the quantitative results for the mmWave experiments in the **Table 2**: <https://anonymous.4open.science/r/ICML2026_Rebuttal-05F5>. RFCanvas achieves the best performance, due to the incorporation of visual priors. Although NeRF$^2$ attains competitive error values, this metric alone can be misleading, as it fails to capture the underlying channel structure. As illustrated in Fig.4 of our paper, NeRF$^2$ is unable to accurately reconstruct the channel structure.
>
> **For Weakness 6:**
>
> Thank you for this valuable feedback. We will carefully revise the manuscript to improve the clarity of the exposition, refine the notation, and enhance the overall readability of the theory section. We would also greatly appreciate it if the reviewer could point out specific parts of the manuscript where the presentation is particularly unclear or difficult to follow. Such guidance would help us better target our revisions and further improve the quality of the paper.
>
> **References:**
>
> [1] Lu, J., Shanbhag, H., \& Al Hassanieh, H. (2025). GeRaF: Neural Geometry Reconstruction from Radio Frequency Signals. In The Thirty-ninth Annual Conference on Neural Information Processing Systems.

---

> > ### Author Rebuttal · Reviewer_fW1g · 2026-04-03
> >
> > Responses provided by the authors to my questions are reasonable and satisfactory. Regarding the presentation, I would suggest focusing on section 4 of the paper to make it more accessible and also summarize the notation in a table. Also, Fig. 1 can come earlier in the intro.

---

> > > ### Author Response · Authors · 2026-04-06
> > >
> > > Thank you for your constructive feedback and helpful suggestions. We are glad that our responses have addressed your questions satisfactorily.
> > >
> > > Following your advice, we will revise the manuscript to improve its presentation. In particular, we will place greater emphasis on Section 4 to enhance clarity and accessibility for readers. We will also include a table summarizing the main notations used throughout the paper for easier reference. Additionally, Fig. 1 will be moved earlier in the introduction to better illustrate the motivation and overall framework.
> > >
> > > We appreciate your insightful comments, which will help us further improve the quality of the paper.

---

### Official Review · Reviewer_adz1 · 2026-03-12

**Soundness:** 2
**Presentation:** 2
**Significance:** 2
**Originality:** 2
**Overall Recommendation:** 3
**Confidence:** 4

**Summary:**

This paper introduces SNRFT, a neural RF channel modeling framework that integrates stochastic transport modeling, implicit scene representations, and differentiable ray tracing. The core concept is to model RF propagation as a stochastic transport process, enabling the learning of material-dependent attenuation while incorporating physical principles such as reciprocity and reversibility. The proposed method explicitly models geometry, material, and phase fields, and is evaluated on tasks including BLE RSSI estimation, mmWave channel estimation, and 5G CSI estimation.

**Compliance With Llm Reviewing Policy:**

Affirmed.

**Key Questions For Authors:**

Could the authors provide ablation studies to isolate the individual contributions of the stochastic transport modeling, the reciprocity/reversibility constraints, and the specific propagation strategies?

How should the applicability boundaries of this method be defined in non-line-of-sight (NLOS) or heavily occluded environments where diffraction and scattering are not negligible?

Can the authors provide visualizations or quantitative validations of the learned geometry, material, and phase fields?

Will the code and datasets be made publicly available? If not, could the paper provide more comprehensive implementation details to ensure reproducibility?

**Limitations:**

The discussion of limitations is currently incomplete. In addition to noting that exponential transport is only an approximation, the paper should more explicitly discuss the limited physical scope of the model, the possible loss of multipath information under the single-branch propagation strategy, the lack of evidence for interpretability, and the current reproducibility limitations.

**Strengths And Weaknesses:**

The primary motivation is to ground neural RF modeling in physics, moving beyond purely black-box approaches. The paper tackles a significant problem with clear relevance to wireless communications, sensing, and localization. The authors' attempt to combine stochastic transport with neural implicit modeling and differentiable ray tracing is an intriguing approach that provides clearer physical motivation than purely black-box alternatives. Furthermore, the evaluation across multiple tasks is a commendable aspect of this work.
However, I find that the current evidence is insufficient to fully support the paper’s primary claims.
-  The physical modeling is incomplete. While the authors motivate the work by describing RF propagation in terms of reflection, diffraction, and scattering, the actual methodology primarily models only specular reflection and penetration.
-  The paper lacks critical ablation studies. Consequently, it is impossible to determine whether the reported performance gains originate from the stochastic transport modeling and associated physical constraints, or simply from the broader neural network architecture and training configurations.
- The claims regarding interpretability lack empirical support at the results level. The paper provides neither visualizations nor quantitative validation for the learned geometry, material, or phase fields.
- Several presentation details undermine my confidence in the paper's technical rigor. These include the imprecise use of physical terminology, loose mathematical formulations, and a lack of clarity in explaining how the scene geometry couples with the electromagnetic observations. Notably, the paper refers to the equation as Snell’s Law, when the expression provided is actually the standard formula for specular reflection, rather than the law of refraction.
- The reproducibility of the paper is limited. The availability of code and data, along with specific implementation details, is not sufficiently articulated.

---

> ### Author Rebuttal · Authors · 2026-03-31
>
> We sincerely thank the reviewer adz1 for the comments and suggestions. We hope the following responses can resolve your questions and concerns.
>
> **For Question 1:**
>
> The stochastic transport modeling and reciprocity/reversibility constraints are inherently coupled in our formulation to derive a closed-form representation of attenuation. Specifically, stochastic transport modeling leads to an underdetermined system of equations (Appendix C, Eq.21-22), and by enforcing reciprocity and reversibility constraints, we obtain a physically consistent closed-form solution. As a result, these components cannot be solely isolated to yield a meaningful representation.
>
> To empirically assess the contribution of this formulation, we compare our theoretical attenuation representation (Eq.14) with an ablated variant where attenuation is directly predicted by naive neural field approach trained with the same architecture and hyperparameters as SNRFT. The results on the BLE dataset across different data densities are summarized in the **Table 1**: <https://anonymous.4open.science/r/ICML2026_Rebuttal-05F5>
>
> For propagation strategies, we follow multipath propagation models [1,2] and traditional ray-tracing frameworks [3], which are well-established in wireless communication and computational graphics.
>
> **For Question 2:**
>
> We emphasize that our framework is built upon a multipath propagation model, which naturally accounts for both LoS and NLoS signal components. Due to the relatively large wavelengths of RF signals (on the order of $10^{-3}$m to $10^{-1}$m), most surfaces are perceived as electrically smooth, and propagation is therefore typically dominated by specular reflections rather than diffuse scattering [4]. Accordingly, our theoretical formulation focuses on specular reflection and penetration. However, the multipath propagation model does take into account the multi-bouncing effects among reflecting surfaces. In addition, our implementation also incorporate first-order diffraction and scattering effects. We acknowledge that this approximation does not fully capture the complexity of diffraction and scattering in highly occluded environments. Extending the current framework to more accurately model these effects remains an important direction for future work. We will further clarify these implementation details and include them under Sec. 6 in the revised manuscript.
>
> **For Question 3:**
>
> We have provided the visualization results in **Figure 1** via the following link:<https://anonymous.4open.science/r/ICML2026_Rebuttal-05F5>. The learned material field and geometry are consistent with the reference image (Fig.4) of the hallway. In particular, the material coefficient remains close to zero in free space, while the learned geometry field takes positive values in free space, which aligns with the expected physical interpretation.
>
> **For Question 4:**
>
> Yes, we will release the code, which is already included in the supplementary materials, and provide additional implementation details in the camera-ready version if it is accepted. Regarding the datasets, as stated in the paper, the BLE and CSI datasets are publicly available. We will also release the mmWave dataset later if the manuscript is accepted.
>
> **For Weakness 4:**
>
> Thanks for your observations. We acknowledge that we only used Snell's law for reflection and not refraction; we will accordingly rephrase as "specular reflection determined by Snell's law''. Please note that we already described how the scene geometry couples with the electromagnetic observations in Section 4.4. However, we are happy to expand the description to further clarify the connections.
>
> **For Weakness 5:**
>
> Thanks for the valuable suggestion. We agree that the current discussion of limitations can be improved. In the revised manuscript, we will expand Section 6 to more clearly discuss the limitations of the exponential transport assumption, as well as the restricted physical scope of the model, including the approximations made for diffraction and scattering effects. We also would like to emphasize again that our framework is based on a multipath propagation model rather than a single-branch approximation. In addition, please note again that we have already included our code in the supplementary materials to facilitate reproducibility.
>
> **References:**
>
> [1] Tse, D., \& Viswanath, P. (2005). Fundamentals of wireless communication. Cambridge university press.
>
> [2] Rappaport, T. S. (2010). Wireless communications: Principles and practice, 2/E. Pearson Education India.
>
> [3] Ling, H., Chou, R. C., \& Lee, S. W. (1989). Shooting and bouncing rays: Calculating the RCS of an arbitrarily shaped cavity. IEEE Transactions on Antennas and propagation, 37(2), 194-205.
>
> [4] Lu, J., Shanbhag, H., \& Al Hassanieh, H. (2025). GeRaF: Neural Geometry Reconstruction from Radio Frequency Signals. In The Thirty-ninth Annual Conference on Neural Information Processing Systems.

---

> > ### Author Rebuttal · Reviewer_adz1 · 2026-04-04
> >
> > The author solved my problem. However, I believe my initial score was reasonable, so I will maintain it.

---

> > > ### Author Response · Authors · 2026-04-06
> > >
> > > We are glad that our response has addressed the concern. If there are any additional aspects where you feel further improvements could strengthen the paper, we would greatly appreciate your insights. Thank you again for your time and thoughtful consideration.

---

### Decision · Program_Chairs · 2026-04-30

**Decision:**

Accept (regular)

**Comment:**

Reviewers appreciated the breadth of the experiments over real-world datasets spanning diverse wireless tasks. The integration of neural representations with physics-based RF propagation modeling was deemed a novel, interesting, and timely contribution.

Many reviewers pointed out the absence of an ablation study as a significant drawback; this, however has been handled in a satisfactory manner during the rebuttal stage.

A lingering concern was reproducibility. It is great that the authors have committed to release the code, but please note that releasing the code is not enough.  The paper is significantly below the bar for what is considered the reproducibility standard for conferences like ICML. The authors are strongly urged to use the supplementary material to describe their datasets, implementation, execution environment, experiment setup (train/test splits, hyperparameter search methodology, etc.)

Please consult the ML reproducibility checklist below for a detailed description of what to include:

https://www.cs.mcgill.ca/~jpineau/ReproducibilityChecklist.pdf

See also:

https://www.jmlr.org/papers/volume22/20-303/20-303.pdf